# Review of electrocardiographic abnormalities among people living with HIV in Sub-Saharan Africa: A systematic review

Andrew Weil Semulimi[1,2]*, Andrew Peter Kyazze[2], Edward Kyalo[3], John Mukisa[4], Charles Batte[1], Felix Bongomin[5], Isaac Ssinabulya[3,6], Bruce J. Kirenga[1,3], Emmy Okello[3,6]

1 Department of Medicine, Lung Institute, College of Health Sciences, Makerere University, Kampala, Uganda, 2 Department of Physiology, School of Biomedical Sciences, College of Health Sciences, Makerere University, Kampala, Uganda, 3 Department of Medicine, School of Medicine, College of Health Sciences, Makerere University, Kampala, Uganda, 4 Department of Immunology and Molecular Biology, School of Biomedical Sciences, College of Health Sciences, Makerere University, Kampala, Uganda, 5 Faculty of Medicine, Department of Medical Microbiology & Immunology, Gulu University, Gulu, Uganda, 6 Division of Adult Cardiology, Uganda Heart Institute, Kampala, Uganda

* andrew.semulimi@mak.ac.ug

**Data Availability Statement:** All relevant data are within the manuscript and its Supporting Information files.

## Abstract

### Introduction

Electrocardiographic (ECG) abnormalities are increasingly being reported among people living with HIV (PLWH). However, the exact prevalence of ECG abnormalities among PLWH in Sub-Saharan Africa (SSA), a region with one of the highest burdens of HIV, is not known. Through a systematic review, we determined the prevalence and patterns of ECG abnormalities among PLWH in SSA.

### Methods

We conducted a search in online databases including EMBASE, MEDLINE, CINAHL and Research for Life for studies published between 1st January 2000 and 31st December 2020. Studies reporting any form of ECG abnormalities published in English were screened and reviewed for eligibility. Retrieved studies were assessed for validity using the modified Newcastle-Ottawa Scale. Data was summarized qualitatively, and ECG abnormalities were further subcategorized into rate, conduction, and rhythm abnormalities as well as atrial and ventricular enlargements.

### Results

We retrieved seventeen of the 219 studies assessed for eligibility published between 2001 and 2020, with a total of 2,572 eligible participants. The mean age of the participants ranged between 6.8 years and 58.6 years. Of the 17 studies, 8 (47%) were case-control, 6 (35.3%) cross-sectional and 3 (17.6%) were cohort in design. Thirteen studies were conducted in the adult population while four were conducted in the pediatric population. The prevalence of ECG abnormalities ranged from 10% to 81% and 6.7% to 26.5% in the adult and pediatric population respectively. Among studies done in the adult population, conduction abnormalities were the most reported (9 studies) with a prevalence ranging from 3.4% to 53.5%. In the

**Funding:** Makerere University Non-Communicable Diseases (MAKNCD) Research Training Program: supported by the Fogarty International Centre of the National Institutes of Health under Award Number D43TW011401.The content is solely the responsibility of the authors and does not necessarily represent the official views of the National Institute of Health. The funders had no role in study design, data collection and analysis, decision to publish, or preparation of the manuscript.

**Competing interests:** The authors have declared that no competing interests exist.

**Abbreviations:** ECG, Electrocardiogram; PLWH, People living with HIV/AIDS; SSA, Sub-Saharan Africa; NOS, Modified Newcastle Ottawa Scale.

pediatric population, rate abnormalities were the most reported (4 studies) with a prevalence ranging from 3.9% to 20.9%. The heterogeneity in results could be attributed to the absence of uniform criteria to define ECG abnormalities.

## Conclusion

Our findings highlight a high prevalence of ECG abnormalities among PLWH in SSA. Consideration of ECG in the comprehensive evaluation of cardiac dysfunction among PLWH in SSA maybe warranted.

## Introduction

Since 1981, close to 80 million people have been infected with HIV, with at least 36 million people dying from HIV-related illnesses [1]. However, improved access to life saving antiretroviral therapy (ART) has led to a significant reduction in HIV- related mortality globally [2]. This has led to an improvement in the life expectancy of people living with HIV (PLWH) [3] predisposing them to non-AIDS-related mortality and morbidity such as cardiovascular diseases (CVDs) [4]. Moreover, the global burden of HIV- related CVDs has almost tripled over the past two decades [5]. In addition, HIV-related CVDs are responsible for about 2.6 million disability adjusted life years which has more than tripled from 0.74 million [6]. It is further estimated that PLWH are 2-fold more likely to develop CVDs than their HIV negative counterparts [6] and by 2030, 78% of PLWH may develop CVD [7].

Sub-Saharan Africa (SSA), which accounts for more than two-thirds (67%) of the global HIV burden has the largest population-attributable fraction of HIV-related CVDs [2, 6]. The high burden of HIV-related CVDs could be attributed to several factors including chronic inflammation [8], low CD4+ T-cell count, high HIV RNA viral load, and longer duration of ART [9, 10] as well as the increasing prevalence of traditional CVD risk factors such as older age, smoking, alcohol use, hypertension, and diabetes mellitus [10]. The rising prevalence of HIV-related CVDs presents a serious conundrum for health workers and undermines global goals to achieve a relative reduction in deaths by non-communicable diseases by 25% by 2025 [11].

Electrocardiogram (ECG) is a non-invasive, and widely available diagnostic modality that can be used in the early identification of CVDs such as arrhythmias and myocardial infarction and can be predictive of incident CVDs [9]. With an increasingly aging population, [12], the prevalence of ECG abnormalities among PLWH is likely to increase in future.

Despite an increase in reporting of ECG abnormalities among PLWH in other regions of the world [9, 13, 14], a precise estimate of the prevalence and patterns of ECG changes among PLWH in SSA is unknown. In this systematic review, we aimed to comprehensively review published data on the prevalence and pattern of ECG abnormalities among PLWH in SSA over a 20-year period.

## Methods

### Study design

A systematic review was conducted in accordance to the Preferred Reporting Items for Systematic Reviews and Meta-Analyses (PRISMA) checklist [15]. **(S1 Checklist)** The protocol was prospectively registered with PROSPERO prior to data extraction (CRD42021243664).

## Search strategy

We developed a search strategy (**S1 Appendix**) with the aid of the Makerere University College of Health Science Sir Albert Cook librarian and a search was conducted through MEDLINE, CINAHL, EMBASE, and Research for Life. A manual search was conducted to identify additional articles. Search terms used were: (("electrocardiogram"[Text Word] OR "cardiac arrhythmias"[Text Word] OR "electrocardio*"[Text Word] OR "12 lead Electrocardiography"[Text Word] OR "EKG"[Text Word] OR "ECG"[Text Word] OR "Long QT Syndrome"[MeSH Terms] OR "arrhythmias, cardiac"[MeSH Terms] OR "Cardiac Conduction System Disease"[MeSH Terms]) **AND** ("HIV"[Text Word] OR "hiv infection*"[Text Word] OR "HIV"[MeSH Terms] OR "HIV Long-Term Survivors"[MeSH Terms]) **AND** ("Sub-Saharan Africa"[Text Word] OR "Africa South of the Sahara"[MeSH Terms])) AND (2000:2020[pdat]).

## Eligibility criteria

Original articles from cross-sectional, case-control, retrospective, prospective, and randomized clinical trial studies, published online from January 2000 to December 2020 reporting any form of ECG abnormality among HIV positive individuals in SSA were included. Systematic reviews, editorials, as well as case reports and case series including less than 10 participants were excluded. In addition, articles published in languages other than English were excluded.

## Study selection process

We found 44 articles from MEDLINE, 125 from Research for Life, 19 from Embase and 46 from CINAHL with 15 duplicates. 219 articles were retrieved and screened for eligibility. However, we failed to retrieve full texts of two articles. 168 articles were excluded by abstract, while 32 were excluded after reviewing the full text. Seventeen articles were included in the review (**Fig 1**). Study titles and/or abstracts collected through the search strategy were screened independently by three authors (EK, AWS, APK) to identify studies that met the inclusion criteria. The eligible articles were coded by two reviewers (AWS, APK) who extracted, further verified, and validated the articles. Any variation that arose was discussed and resolved by a third reviewer (FB).

## Data extraction process

A data extraction sheet was designed using Microsoft Excel V.2016 and used to extract the data from the articles. For each article, the following information was extracted: first author, year of publication, country where the study was conducted, study design, age (mean or median), gender characteristics, the sample size, prevalence of ECG abnormalities, criteria used to define the ECG abnormality, how the diagnosis was made, and the period of data collection.

## Quality assessment

The modified Newcastle Ottawa scale (NOS) [16] was used to assess for the risk of bias. Eligible studies with a score between 7 and 9 were considered to have a low risk of bias, while those with a score between 4 and 6 were considered to have a high risk of bias. Studies that scored between 0 and 3 were considered to have a very high risk of bias (**S2 Checklist**).

## Data analysis

The data was analyzed and summarized qualitatively based on the different ECG changes which included: rate abnormalities, rhythm abnormalities, conduction abnormalities, ST

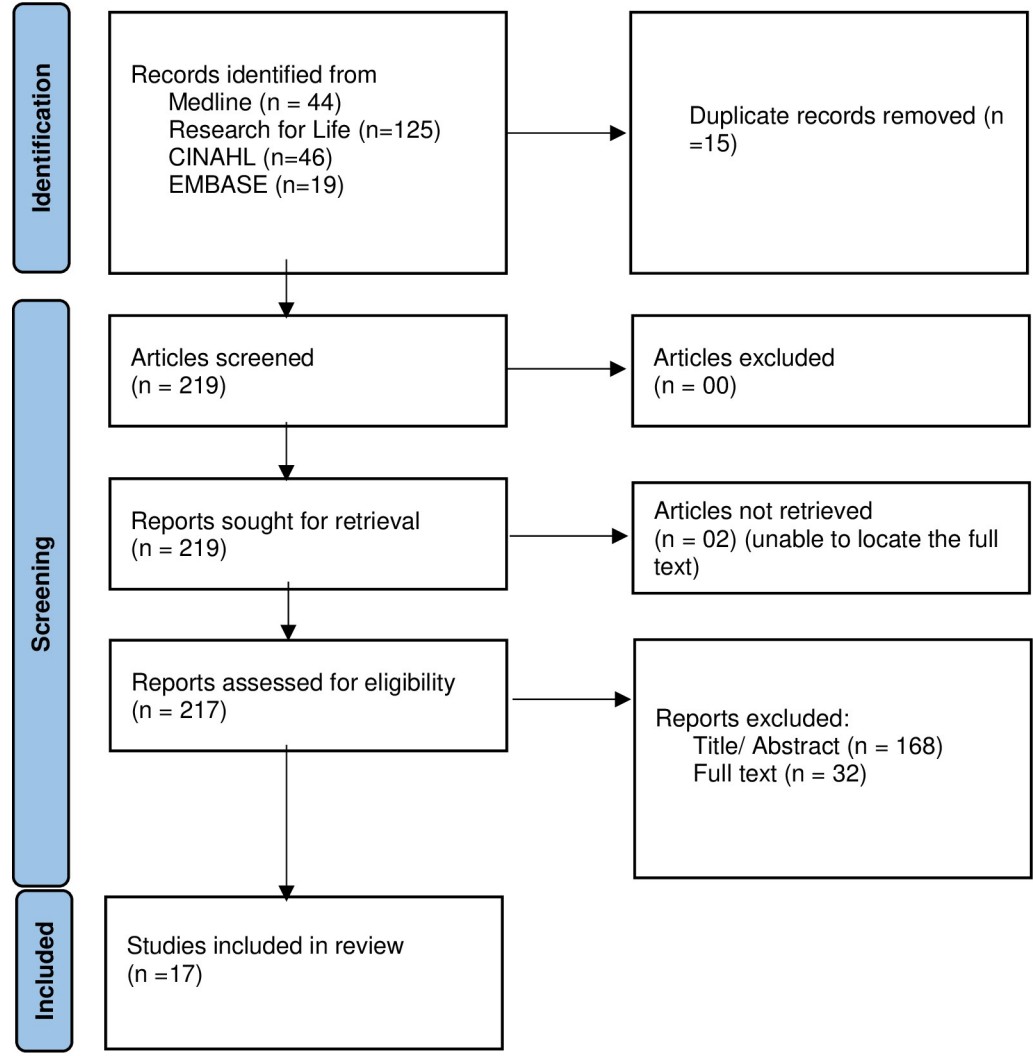

**Fig 1. PRISMA flow diagram.**

changes, ischemia/ infarction, and atrial or ventricular enlargement. A formal meta-analysis was not possible due to the variability in study designs and reported findings.

## Results

### Study characteristics

Of the 219 studies screened for eligibility, 17 studies [17–33] which recruited a total of 2,572 PWLH were included in the final analysis (**Fig 1**). Four of the selected studies were conducted in the pediatric population [18, 25, 28, 33] while 13 of the seventeen were conducted in the adult population [17, 19–24, 26, 27, 30–32]. Regarding the study design, 8 (47%) were case-control studies [18, 20, 21, 23, 24, 29, 31, 33] while 6 (35.3%) and 3 (17.6%) were cross-sectional [17, 25–28, 32] and cohort [19, 22, 30] studies, respectively. The studies were conducted in 7 African countries with majority (n = 8) done in Nigeria [17–19, 24, 27, 31, 32]. In addition, we found one multicenter study with three countries involved [19]. In the reviewed studies, the most reported definition of ECG abnormalities was QTc interval prolongation defined as

more than 0.44 seconds (n = 4) [22, 25, 27, 33]. Regarding ST changes, ST elevations were the most reported (n = 4). Studies were carried out between 1997 and 2016 and published between 2001 and 2020. The mean age of the adult participants ranged between 32±10.5 and 48±13.1 years (**Table 1: Studies reporting ECG abnormalities in adult PLWH in SSA**) while that of the pediatric participants was between 8.30±3.92 and 8.41±3.99 years (**Table 2: Studies reporting ECG abnormalities in pediatric PLWH in SSA**).

## ECG abnormalities

From the 13 eligible studies conducted among adult PLWH (**Table 3**), the prevalence of ECG abnormalities ranged from 10% to 81% [20, 21, 24, 27, 30–32] while in the studies conducted in children, the prevalence of ECG abnormalities was between 6.7% and 26.5% [18, 25, 28].

## ECG abnormalities among adult PLWH

Regarding the different types of abnormalities, conduction abnormalities were the most reported. Nine studies reported different forms of conduction abnormalities [17, 19, 20, 22–24, 26, 27, 31] with a prevalence ranging between 3.4% and 53.3%. The most common

**Table 1. Studies reporting ECG abnormalities in adult PLWH in SSA.**

| First Author (year) | Study title | Study Country | Study design | Sample size and study population characteristics | Age | Female, (%) | Period of data collection |
|---|---|---|---|---|---|---|---|
| Hamadou B (2017) [23] | Echocardiographic and electrocardiographic abnormalities in adults living with human immunodeficiency virus: a cross-sectional study in the Yaoundé Central Hospital, Cameroon | Cameroon | case-control study | PLWH, n = 59 | 47±12.7 | 67.8 | 3 months (March to May 2016) |
| Menanga AP (2015) [26] | Patterns of cardiovascular disease in a group of HIV-infected adults in Yaoundé, Cameroon. | | cross-sectional study | 53 | 48±13.1 | 52 | February— July 2014 |
| | | | | Average CD4 cell counts/ mm³, 205±187.9 | | | |
| | | | | Average duration on ART (months), 33.7±39.3. | | | |
| | | | | WHO stage I, 6 (13.6) | | | |
| | | | | WHO stage II, 8 (18.2) | | | |
| | | | | WHO stage III, 11 (25.0) | | | |
| | | | | WHO stage IV, 19 (43.2) | | | |
| Berhe N (2001) [30] | Electrocardiographic findings in Ethiopians on pentavalent antimony therapy for visceral leishmaniasis | Ethiopia | Cohort | PLWH, n = 10 | 29.3 (6.3) | 22.4 | August 1994— December 1997 |
| | | | | HIV-negative, n = 39 | 20.9 (10.6) | | |
| | | | | | 16.5 (63) | | |
| Appiah LT (2019) [58] | Cardiovascular risk factors among Ghanaian patients with HIV: A cross-sectional study | Ghana | Case control | PLWH, n = 345 | 41 ± 11 | 72 | January 2013 —May 2014 |
| | | | | PLWH: CD4 count, Median (IQR) total absolute lymphocyte CD4 T cell counts, 323 [120, 536] cells/ μL. | | | |
| | | | | PLWH on ART (nevirapine- or efavirenz-based regimen), n = 173 | | | |
| | | | | PLWH on ART: Median (IQR) total absolute lymphocyte CD4 T cell counts, 164 [56, 290] cells/μL | | | |
| | | | | HIV negative, n = 161 | 32 ± 10.5 | 28 | |

*(Continued)*

**Table 1.** (Continued)

| First Author (year) | Study title | Study Country | Study design | Sample size and study population characteristics | Age | Female, (%) | Period of data collection |
|---|---|---|---|---|---|---|---|
| Kumwenda JJ (2005) [29] | Differential diagnosis of stroke in a setting of high HIV prevalence in Blantyre, Malawi | Malawi | Case control | PLWH, n = 47 | 37.5 (13.1) | 59.6 | October 2001—July 2002 |
| | | | | HIV-negative, n = 51 | 58.6 (16.8) | 41.1 | |
| Ogunmodede JA (2017) [24] | Structural echocardiographic abnormalities seen in HIV/AIDS patients are independent of CD4 count | Nigeria | Case control | PLWH, n = 150 | 37.3 ± 8.9 | 57.3 | |
| | | | | HIV negative, n = 150 | 40.1 ± 16.9 | 57.3 | |
| Okoye (2017) [31] | Electrocardiographic abnormalities in treatment-naïve HIV subjects in south-east Nigeria | | Case Control | PLWH Treatment naive, n = 250 | 34.89± 10.58 | 49.6% | September to December, 2015 |
| | | | | HIV negative, n = 200 | 36.04±12.61 | | |
| Njoku PO (2016) [17] | Electrocardiographic findings in a cross-sectional study of human immunodeficiency virus (HIV) patients in Enugu, south-east Nigeria | | cross-sectional study | HIV-infected on ART, n = 100 | 35.85 ± 8.94 | 49 | November 2010—November 2011 |
| | | | | Median CD4, 137 inter quartile range (IQR), 46–246) | | | |
| | | | | HIV-infected ART-naïve, n = 100 | 34.43 ± 9.49 | 52 | |
| | | | | HIV-negative controls, n = 100 | 35.76 ± 9.74 | 48 | |
| Isiguzo G (2013) [32] | Determinants of HIV-related cardiac disease among adults in north central Nigeria | | Cross-sectional study | 200 | 37±9 | 71 | |
| | | | | On ART, 168 (84.4%) | | | |
| | | | | Median Duration of ART, 46 months (IQR 18–64 months) | | | |
| | | | | WHO stage I, 26/81 (31%) | | | |
| | | | | WHO stage II, 20/81 (28.8) | | | |
| | | | | WHO stage III, 30 (35.7) | | | |
| | | | | WHO stage IV, 8/81 (9.5) | | | |
| | | | | CD4 cell count > 200, 51 (25.7%) | | | |
| | | | | $Log_{10}$ viral load of those with cardiac disease, mean (SD), 3.16 (1.1) | | | |
| | | | | $Log_{10}$ viral load of those without cardiac disease, mean (SD), 3.22 (1.1) | | | |
| Sani MU (2005) [27] | QTc interval prolongation in patients with HIV and AIDS | | Cross-sectional study | AIDS patients, n = 100 | 34.2 ± 8.9 | 51 | January to October 2001 |
| | | | | PLWH, n = 78 | 33.8 ± 8.9 | 48.7 | |
| | | | | HIV negative, n = 80 | 34.1 ± 8.7 | 47.5 | |
| Hughes J (2019) [22] | Adverse events among people on delamanid for rifampicin-resistant tuberculosis in a high HIV prevalence setting | South Africa | Cohort study | PLWH, n = 46 | 35 (30–43) | 48 | 13 months |
| | | | | CD4 count at DLM start, cells/mm3, median [IQR], 173 [70–294] | | | |
| | | | | On ARVs before DLM, n (% of HIV-positive), n = 38 (83) | | | |
| | | | | ART Regimen: EFV and NRTI 26 (68) | | | |
| | | | | NVP and NRTIs, 6 (16) | | | |
| | | | | Ritonavir plus ATZ/ lopinavir and NRTIs, N = 6 (16) | | | |
| | | | | HIV Negative, n = 12 | 32 (18–39) | 33 | |

(*Continued*)

**Table 1.** (Continued)

| First Author (year) | Study title | Study Country | Study design | Sample size and study population characteristics | Age | Female, (%) | Period of data collection |
|---|---|---|---|---|---|---|---|
| Kentoffio K (2019) [20] | Electrocardiographic Evidence of Cardiac Disease by Sex and HIV Serostatus in Mbarara, Uganda | Uganda | Case control | Age-Sex matched PLWH, n = 59 | 51.2 (6.6) | 47.7 | |
| | | | | Undetectable VL (n, %) = 124 (82.7%) | | | |
| | | | | Age and sex matched HIV negative, n = 154 | 51.4 (7.8) | 50 | |
| Mayosi BM (2006) [19] | Clinical characteristics and initial management of patients with tuberculous pericarditis in the HIV era: the Investigation of the Management of Pericarditis in Africa (IMPI Africa) registry | Cameroon, Nigeria, South Africa | Cohort | Clinical HIV, n = 74 | 36 (18–87) | 39 | 01 March 2004-31st October, 2004 |
| | | | | No clinical HIV disease, n = 111 | 32 (15–79) | 61 | |

conduction abnormality was QTc prolongation which was reported in five studies with a prevalence ranging from 34.7% to 48% [20, 22, 24, 27, 31]. In two of these studies, QTc prolongation was defined as an interval of more than 0.44 seconds [22, 27].

**Table 2. Studies reporting ECG abnormalities in pediatric PLWH in SSA.**

| First Author (year) | Study title | Study Country | Study design | Sample size and population characteristics | Age | Female, (%) | Period of data collection |
|---|---|---|---|---|---|---|---|
| Attamah CA (2020) [18] | Electrocardiographic findings in human immunodeficiency virus-infected children in Benin City, Nigeria | Nigeria | Case control | PLWH, n = 200 | 8.30 ± 3.92 | 38 | January-June 2015 |
| | | | | Age matched HIV negative, n = 200 | 8.41 ± 3.99 years | 40 | |
| Ige O (2014) [33] | The QT interval in human immunodeficiency virus-positive Nigerian children | | Case-control | PLWH, n = 100 | 6.53 (5.81, 7.25) | | 2008 |
| | | | | WHO stage I, 69 (69%) | | | |
| | | | | WHO stage II, 14 (14%) | | | |
| | | | | WHO stage III, 17 (17%) | | | |
| | | | | ART, 62 (62%) | | | |
| | | | | No significant immunological deficiency, 57 (57%) | | | |
| | | | | Mild immunodeficiency, 10 (10) | | | |
| | | | | Advanced, 8 (8%) | | | |
| | | | | Severe, 25 (25%) | | | |
| | | | | HIV Negative, n = 100 | 6.53 (5.80, 7.26) | | |
| Namuyonga J (2016) [25] | Cardiac Dysfunction Among Ugandan HIV-infected Children on Antiretroviral Therapy | Uganda | Cross-sectional | 285 | 9 (6, 13) | 54 | July 2012—January 2013 |
| | | | | ART Duration <5, 166 (58) | | | |
| | | | | ART duration, ≥ 5, 119 (42) | | | |
| | | | | Viral load (copies/ml) (n = 217), 20 (20,76) | | | |
| | | | | CD4 count (cells/mm$^3$), 944, (596,3462) | | | |
| | | | | Virological Suppression (<400 copies/ml), 194 (194) | | | |
| Lubega S (2005) [28] | Heart disease among children with HIV/AIDS attending the pediatric infectious disease clinic at Mulago Hospital | | Cross-sectional study | 230 | 6.8 years (range 10 months to 16 years, SD = 3.6 years) | 51.3 | September 2002—February 2003 |

**Table 3. Adult PLWH in the reviewed studies with ECG abnormalities.**

| First Author | Diagnosis of ECG abnormality N (%) | ECG changes | | | | | |
|---|---|---|---|---|---|---|---|
| | | Rate Abnormalities, N (%) | Rhythm Abnormalities, N (%) | Conduction Abnormalities, N (%) | ST changes, N (%) | Ischemia/Infarction, N (%) | Atrial and Ventricular enlargement, N (%) |
| Hamadou B (2017) [23] | | Sinus tachycardia, n = 6 (10.2) | Atrial fibrillation, n = 3 (5.1) | Abnormal repolarization, n = 7 (11.9) | ST segment elevation, n = 1 (1.7) | Q waves, n = 1 (1.7) | Left Atrial enlargement, n = 1 (1.7) |
| | | Sinus bradycardia, n = 7 (11.9) | | Premature contractions, n = 2 (3.4) | Isolated T waves, n = 1 (1.7) | | Left Ventricular hypertrophy, n = 2 (3.40) |
| | | Ventricular Tachycardia, n = 2 (3.4) | | | | | |
| Menanga AP (2015) [26] | | Sinus tachycardia, n = 25 (56.8) | Arrhythmias, n = 9 (20.4) | Abnormal repolarization (59%, n = 26) | | | Left Ventricular hypertrophy, n = 13 (29.5) |
| | | | | Conduction anomalies, n = 11 (25.0) | | | Right Ventricular Hypertrophy, n = 3 (6.8) |
| | | | | Low voltage, n = 8 (18.1) | | | Right Atrial Enlargement, n = 3 (6.8) |
| | | | | Slow progression of the R waves, n = 3 (6.8) | | | Left Atrial Enlargement, n = 1 (2.3) |
| Berhe N (2001) [30] | New ECG changes, n = 2 (20%) | | | | | | |
| Appiah LT (2019) [58] | Major ECG abnormalities, n = 162 (47) | | | | | | |
| | Minor ECG abnormalities, n = 35 (10) | | | | | | |
| Kumwenda JJ (2005) [29] | | | | | | | Left Ventricular Hypertrophy, n = 17 (36) |
| Ogunmodede JA (2017) [24] | Abnormal ECG, n = 79 (53) | | | First Degree AV Block, n = 5, (53.3) | | | Left Atrial Enlargement, n = 30 (20) |
| | | | | Premature Ventricular Contractions, n = 1 (8) | | | Right Atrial Enlargement, n = 11 (7.3) |
| | | | | QTc prolongation, n = 52 (34.7) | | | Left Ventricular hypertrophy, n = 26 (17.3) |
| | | | | | | | Right ventricular hypertrophy, n = 13 (8.7) |

*(Continued)*

**Table 3.** (Continued)

| First Author | Diagnosis of ECG abnormality N (%) | Rate Abnormalities, N (%) | Rhythm Abnormalities, N (%) | Conduction Abnormalities, N (%) | ST changes, N (%) | Ischemia/ Infarction, N (%) | Atrial and Ventricular enlargement, N (%) |
|---|---|---|---|---|---|---|---|
| | | **ECG changes** | | | | | |
| Okoye (2017) [31] | ECG abnormalities, n = 175 (70%) | Sinus Tachycardia, n = 160 (64); Sinus Bradycardia, n = 2 (0.8) | Ventricular ectopic, n = 10 (4); Atrial Ectopic, n = 2 (0.8) | Prolonged QTc, n = 120 (48); Shortened PR interval, n = 2 (0.8); Left Axis deviation, n = 4 (1.6); First Degree heart block, n = 6 (2.4); Left anterior hemiblock, n = 2 (0.8); Incomplete Right bundle branch block, n = 4 (1.6); Low QRS in all leads, n = 10 (4); Low QRS in limb leads, n = 8 (3.2) | ST depression, n = 75 (30); T wave inversion, n = 54 (21.6) | | Left ventricular hypertrophy, n = 35 (14); Right Ventricular Hypertrophy, n = 2 (0.8) |
| Njoku PO (2016) [17] | | **PLWH on ART** / **PLWH ART naive**: Sinus Tachycardia, n = 8(8.6) / Sinus Tachycardia, n = 14 (19.2) | | **PLWH on ART** / **PLWH ART naive**: Left Bundle Branch block, n = 1 (1.1) / Left Bundle branch block, n = 00; Right Bundle Branch block, n = 1 (1.1) / —; Ventricular Ectopic beats, n = 1 (1.1) / Ventricular Ectopic beats, n = 1 (1.1); First Degree Heart block, n = 3 (3.2) / First Degree Heart block, n = 1 (1.4); Left Axis Deviation, n = 15 (16) / Left Axis Deviation, n = 10 (13.7) | **PLWH on ART** / **PLWH ART naive**: — / ST-segment elevation, n = 2 (2.7); T-wave inversion in leads II, III, aVF (inferior leads), n = 2 (2.2) / T-wave inversion in leads II, III, aVF (inferior leads), n = 1 (1.4); — / T-wave inversion in leads I, aVL, V5–V6 (lateral leads), n = 2 (2.7); T wave inversion in leads V1 – V3, n = 44 (47) / T wave inversion in leads V1 – V3, 22 (30.4) | | **PLWH on ART** / **PLWH ART naive**: — / Left Ventricular Hypertrophy, n = 8 (11) |

*(Continued)*

**Table 3.** (Continued)

| First Author | Diagnosis of ECG abnormality N (%) | ECG changes | | Rhythm Abnormalities, N (%) | Conduction Abnormalities, N (%) | ST changes, N (%) | Ischemia/ Infarction, N (%) | Atrial and Ventricular enlargement, N (%) |
|---|---|---|---|---|---|---|---|---|
| | | Rate Abnormalities, N (%) | | | | | | |
| Isiguzo G (2013) [32] | ECG abnormalities, n = (57.3) | Bradycardia, n = 6 (3); Tachycardia, n = 10 (5) | | Arrhythmias, n = 8 (4) | First Degree Heart block, n = 6 (3); Low Voltage Complex, n = 3 (1.5); Right Bundle Branch Block, n = 1 (0.5); Right Ventricular Strain, n = 6 (3) | | | Left Atrial Enlargement, n = 2 (1); Left Ventricular hypertrophy, n = 58 (29); Left Ventricular hypertrophy and bradycardia, n = 9 (4.5); Left Ventricular hypertrophy and premature ventricular contraction, n = 1 (0.5); Right Ventricular hypertrophy, n = 1 (0.5) |
| Sani MU (2005) [27] | AIDS: Abnormal ECG, n = 81 (81%); HIV: Abnormal ECG, n = 51 (61%) | | | | AIDS: QTc Prolongation, n = 45 (45%) | HIV: QTc Prolongation, n = 22 (28%) | | |
| Hughes J (2019) [22] | | | | | QT interval prolongation, n = 13 (28) | | | |
| Kentoffio K (2019) [20] | ≥ 1 ECG abnormality, n = 32 (20.7) | | | Atrial fibrillation, n = 00 | Inter-ventricular conduction delay, n = 9 (5.8); Right Bundle branch block, n = 4 (2.6); QTc prolongation, n = 4 (2.6); Left bundle branch, n = 2 (1.3); L-axis deviation, n = 2 (1.3); R-axis deviation, n = 1 (0.7) | ST depressions, n = 01 (0.7) | Ischemic ECG, n = 14 (9.0); Q waves, n = 2 (1.3); Ischemic ST depressions or T-wave inversions, n = 12 (7.7) | Left ventricular hypertrophy, n = 8 (5.2); Left atrial abnormality, n = 11 (7.1); Right Ventricular hypertrophy, n = 2 (1.3) |
| Mayosi BM (2006) [19] | | | | Atrial fibrillation, n = 2 (16.7) | Electrical alternans n = 5 (45.5); Micro voltage, n = 8 (29.6); PR Segment elevation, n = 11 (55) | ST-segment elevation, n = 14 (58.3) | | |

ART –Antiretroviral Therapy

**Table 4. Pediatric PLWH in the reviewed studies with ECG abnormalities.**

| First Author | Diagnosis of ECG abnormality N (%) | ECG changes | | | | | |
| --- | --- | --- | --- | --- | --- | --- | --- |
| | | Rate Abnormalities, N (%) | Rhythm Abnormalities, N (%) | Conduction Abnormalities, N (%) | ST changes, N (%) | Ischemia/ Infarction, N (%) | Atrial and Ventricular enlargement, N (%) |
| Attamah CA (2020) [18] | ECG changes, n = 63 (34.5)<br><br>Multiple ECG changes, n = 22 (11) | Sinus Tachycardia, n = 37 (18.5) | | First degree heart block, n = 10 (5.0)<br><br>Prolonged QRS interval, n = 7 (3.5)<br><br>Prolonged QTc interval, n = 7, (3.5) | ST-segment changes n = 17, (8.5) | ST elevation, n = 4(2.0) | Right Ventricular hypertrophy, n = 13 (6.5)<br><br>Left Atrial Hypertrophy, n = 14 (7)<br><br>Left Ventricular Hypertrophy, n = 17 (8.5)<br><br>Biventricular hypertrophy, n = 1 (0.5) |
| Ige O (2014) [33] | | Sinus Tachycardia, n = 9 (9) | | QTc prolongation, n = 18 (18) | | | |
| Namuyonga J (2016) [25] | ECG abnormalities, n = 19 (6.7) | Sinus tachycardia n = 11 (3.9) | | First degree heart block, n = 1, (0.4)<br><br>Prolonged QTC interval, n = 4, (1.4) | Non-specific T wave changes, n = 13 (4.6) | | Left Ventricular hypertrophy, n = 14 (4.9%)<br><br>Right Atrial Enlargement, n = 1 (0.4) |
| Lubega S (2005) [28] | ECG abnormalities, n = 61 (26.5%) | Sinus tachycardia, n = 48 (20.9) | Ventricular ectopic beats, n = 1 (0.4) | Partial right bundle branch, n = 6 (2.6) | | | Right Ventricular Hypertrophy, n = 6 (2.6)<br><br>Left Ventricular hypertrophy, n = 3 (1.3%) |

ART–Antiretroviral Therapy

Atrial and ventricular enlargement were reported in eight studies [17, 20, 23, 24, 26, 29, 31, 32]. The prevalence of atrial and ventricular enlargement was 1.7% and 29%. Additionally, all eight studies reported prevalence of left ventricular hypertrophy which ranged from 0.5% to 29% [17, 20, 23, 24, 26, 29, 31, 32].

Five studies reported rate abnormalities with a prevalence ranging from 0.8% to 64% [17, 23, 26, 31, 32]. Among the rate abnormalities, the prevalence of sinus tachycardia, was between 8.6% and 64%. Sinus tachycardia was the most common rate abnormality reported in all five studies [17, 23, 26, 31, 32].

ST changes were reported in five studies with a prevalence of 0.7%—58.3% [17, 19, 20, 23, 31]. Of the 5 studies, 3 (43%) studies reported ST elevation whose prevalence was 1.7% to 58.3% [17, 19, 23]. Ischemic changes were reported in 2 studies with a prevalence ranging from 1.7% to 7.7% [20, 23].

The prevalence of arrhythmias was 0.4%—20.4%, and this was reported in five studies [19, 23, 26, 31, 32]. Two of the 5 studies (33%) reported the prevalence of atrial fibrillation that ranged from 5.1% to 16.7% [19, 23].

## ECG abnormalities among children with HIV

Table 4 describes pediatric PLWH in the reviewed studies with ECG abnormalities. Rate abnormalities were the most commonly reported ECG abnormalities (n = 4) [18, 25, 28, 33],

followed by atrial and ventricular enlargement (n = 3), and lastly conduction abnormalities (n = 3) [18, 25, 28]. Regarding rate abnormalities, the prevalence of sinus tachycardia was between 3.9% to 20.9% [18, 25, 28, 33]. Atrial and ventricular enlargements were reported in three studies [18, 25, 28]. Left ventricular hypertrophy determined using Sokolow Index SV (SV1 + RV5/6 ≥35 mm) was the most common atrial and ventricular enlargement reported with a prevalence of 1.3% to 8.5% [18, 25, 28]. QTc prolongation, with prevalence of 1.4% to 18%, was the most reported conduction abnormalities [18, 25, 28].

## Discussion

Although it has been reported that the burden of CVDs among PLWH have increased significantly [5, 6, 34], the burden of these diseases among PLWH in SSA is largely unknown. Our systematic review provides a comprehensive review of published ECG abnormalities among PLWH in SSA. In these studies, ECG abnormalities ranged between 10% to 81% among adults and 6.7%—26.5% in children. In addition, conduction abnormalities were the most reported ECG abnormality among adults while rate abnormalities were the most common ECG abnormality among children.

Several studies have investigated on the prevalence of ECG changes in PLWH. The prevalence of ECG abnormalities among adults was similar to what has been reported in high income countries [9, 13, 35, 36]. In the Strategies for Management of Antiretroviral Therapy (SMART) study which recruited children (>13 years) and adults from 23 different high income counties [37], 19%—51% of participants were found to have either a minor or major ECG abnormality with close to half (49%) having minor abnormalities [9, 36].

The wide range of ECG abnormalities could be attributed to the heterogeneity of the results and the differences in the study population. Regarding the discrepancies in the ECG reporting, several criteria such as American Heart Association Standard criteria [38], normal ECG standard for infants and children [39] and Minnesota Coding system for ECG classification [40] were used by the different studies which may have influenced the prevalence reported. In addition, the characteristics of participants included in the reviewed studies could have contributed to the reported wide range. For instance, Okoye and colleagues [31] studied PLWH who were ART naïve, aged 15 years and above with 30% of their study participants having a low CD4 count (<200 cells/μl) while in a cross-sectional study conducted in Ghana [21], most of the participants had stage III severity of disease with more than half of them (194, 56.3%) having >1 CVD risk. Among the studies conducted in the pediatric population, factors such as wasting [25] and diseases severity [28] influenced results contributing to a wider range in the prevalence of ECG abnormalities. Therefore, the absence of a uniform and standardized tool that can be adjusted for patient characteristics calls for the development of such tools to classify ECG abnormalities. This could inform risk stratification of PLWH at a risk of developing CVDs.

In this study, we found that prevalence of QTc prolongation in adults was the most common conduction abnormality. Several studies done in other settings have showed that the prevalence of QTc prolongation in adults is high [41–45] which is consistent with our findings. The HIV effects on cardiomyocytes leading to fibrosis as well as certain classes of ART such as non-nucleoside reverse transcriptase inhibitors which have been part of the first line regimen until recently could have contributed to the high prevalence of conduction abnormalities and QTc prolongation [41, 46]. This is worrisome since QTc prolongation is an independent risk for ventricular arrhythmias and sudden cardiac death [47].

Regarding atrial and ventricular enlargement in adults, findings from our study are similar to those of a cross-sectional study done in China, which reported a prevalence of left

ventricular hypertrophy of 6.8% [13]. In children, left ventricular hypertrophy was the second most common ECG abnormality. HIV and opportunistic infections coupled with other risk factors of CVDs such as smoking cause inflammation and release of cytokines which have been shown to predispose PLWH to atrial and ventricular hypertrophy [48]. This increases the likelihood of individuals to developing ventricular and atrial arrhythmias [49], congestive heart failure and ischemic heart disease [50] which significantly contribute to CVD related mortality and morbidity.

In the pediatric population, sinus tachycardia, a rate abnormality was the most common ECG abnormality among children unlike in the adult population. This was lower than what was reported in the United States of America (USA) [51] which was conducted before the introduction of combined ART. The higher prevalence in the USA study could be attributed to the increased risk of children to opportunistic infections subsequently leading to CVDs [52]. HIV instigates viral induced myocarditis which is one of the drivers of sinus tachycardia [53] among PLWH. Sinus tachycardia predisposes patients to life threatening arrhythmias, sudden cardiac death and myocardial infarction and its presence necessitates frequent cardiac monitoring.

From the reviewed studies, atrial fibrillation is the most reported arrhythmia which is concordant with results from other large epidemiological studies [13, 54, 55]. Studies have shown a relationship between HIV and atrial fibrillation although the underlying mechanism has not been fully elucidated [56]. Atrial fibrillation causes disruptions in atrial filling and emptying leading to poor clinical outcomes such as cerebrovascular accidents, decompensated congestive cardiac failure, ischemia and sudden cardiac death causing significant morbidity and mortality in this population [29]. This highlights how essential ECGs are in diagnosing PLWH with existing cardiac abnormality and dysfunction if used routinely. However, in SSA, the routine use of PLWH to ECGs is marred by the cost and the need to train health workers to read and analyse them [57].

Our study had several limitations. We limited our study to studies written in English which may have led to the exclusion of other studies reporting ECG abnormalities among PLWH in SSA. Additionally, there is no uniform criteria to define ECG abnormalities which likely resulted in significant heterogeneity in results. Studies by Hughes [22] and Mayosi [19] as well as Berhe [30] were conducted among PLWH who had multi drug tuberculosis, tuberculosis pericarditis and visceral leishmaniasis respectively which may have been associated or their synergistic effect with HIV could have led to the development of ECG abnormalities.

Despite the limitations, our review is the first comprehensive review detailing ECG abnormalities among PLWH in SSA and could be used to depict the burden of cardiovascular diseases among PLWH in SSA. Additionally, we considered a large sample size and various data bases.

## Future direction

Due to the high prevalence of ECG abnormalities among PLWH, more empirical studies should be done in this population. Additionally, longitudinal, and mechanistic studies should be done to explore the casual relationship between HIV and abnormal ECG findings.

## Conclusion

In our review, the ECG abnormalities were present in a significant number of both adult and pediatric reviewed studies. Among adults, conduction disorders were the most common ECG abnormalities reported while in pediatric, rate abnormalities were the most reported ECG abnormality. ECG should be incorporate in the comprehensive cardiovascular risk assessment of PLWH in SSA.

## Supporting information

**S1 Checklist. PRISMA checklist.**
(DOCX)

**S2 Checklist. Newcastle Ottawa scale.**
(XLSX)

**S1 Appendix. Search strategy.**
(DOCX)

## Acknowledgments

Special thanks to Dr. Alison Annet Kinengyere and the Africa Centre for Systematic Reviews and Knowledge Translation at the College of Health Sciences, Makerere University for the assistance rendered to the team in developing the search strategy.

## Author Contributions

**Conceptualization:** Andrew Weil Semulimi, Andrew Peter Kyazze, Charles Batte, Felix Bongomin.

**Data curation:** Andrew Weil Semulimi, Andrew Peter Kyazze, Edward Kyalo, John Mukisa, Felix Bongomin, Emmy Okello.

**Formal analysis:** Andrew Weil Semulimi, Andrew Peter Kyazze, Edward Kyalo, John Mukisa, Felix Bongomin, Isaac Ssinabulya, Emmy Okello.

**Funding acquisition:** Andrew Weil Semulimi, Charles Batte, Felix Bongomin, Bruce J. Kirenga.

**Methodology:** Andrew Weil Semulimi, Andrew Peter Kyazze, Edward Kyalo, John Mukisa, Charles Batte, Felix Bongomin, Isaac Ssinabulya, Bruce J. Kirenga.

**Supervision:** Andrew Weil Semulimi, Felix Bongomin, Bruce J. Kirenga.

**Validation:** Andrew Peter Kyazze, Edward Kyalo, John Mukisa, Felix Bongomin.

**Writing – original draft:** Andrew Weil Semulimi, Andrew Peter Kyazze, Edward Kyalo, John Mukisa, Charles Batte, Felix Bongomin, Isaac Ssinabulya.

**Writing – review & editing:** Andrew Weil Semulimi, Andrew Peter Kyazze, Edward Kyalo, John Mukisa, Charles Batte, Felix Bongomin, Isaac Ssinabulya, Bruce J. Kirenga, Emmy Okello.

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
