## [Decision Letter · Decision Letter 0]

22 Nov 2021

PONE-D-21-34132PREVALENCE OF ELECTROCARDIOGRAPHIC ABNORMALITIES AMONG PEOPLE LIVING WITH HIV IN SUB-SAHARAN AFRICA: A SYSTEMATIC REVIEWPLOS ONE

Dear Dr. Semulimi,

Thank you for submitting your manuscript to PLOS ONE. After careful consideration, we feel that it has merit but does not fully meet PLOS ONE’s publication criteria as it currently stands. Therefore, we invite you to submit a revised version of the manuscript that addresses the points raised during the review process.

I believe the reviewers provided a fair and thorough review of your manuscript, and I encourage you to carefully consider their comments, paying particular attention to areas where the writing is unclear and addressing the heterogeneity of studies analyzed and the limitations these bring (such as including children and adults, and including studies of patients with TB pericarditis). I agree with the reviewers that the range of ECG abnormalities is so large that it is difficult to understand their overall significance. In addition to thoroughly addressing each of the comments of the reviewers, I encourage the authors to focus on which abnormalities are considered to be most clinically significant.

We look forward to receiving your revised manuscript.

Kind regards,

Nathan Maclyn Thielman, MD, MPH

Academic Editor

PLOS ONE

Journal Requirements:

2. Please remove all personal information, ensure that the data shared are in accordance with participant consent, and re-upload a fully anonymized data set. 

Reviewers' comments:

Reviewer's Responses to Questions

**Comments to the Author**

1. Is the manuscript technically sound, and do the data support the conclusions?

Reviewer #1: Yes

Reviewer #2: Yes

2. Has the statistical analysis been performed appropriately and rigorously? 

Reviewer #1: Yes

Reviewer #2: Yes

3. Have the authors made all data underlying the findings in their manuscript fully available?

Reviewer #1: Yes

Reviewer #2: Yes

4. Is the manuscript presented in an intelligible fashion and written in standard English?

Reviewer #1: Yes

Reviewer #2: Yes

5. Review Comments to the Author

Reviewer #1: Overall this is a clear and well-written manuscript synthesizing data about a problem that has been under-studied. Here are my minor comments:

Intro:

"and is highly predictive of incident CVDs (9, 12)"--I do not think this statement is true or justified by the references. Consider rewording--"and can be predictive of incident CVDs"

"Moreover, ECG abnormalities are highly prevalent in the general population and HIV population with the

prevalence of ECG abnormalities ranging from 1.04% to 51% (9, 13-18)." I do not understand this sentence--you seem to be citing several articles about HIV-infected and HIV-uninfected patients in high-income and low-income settings. Overall, I do not think this sentence adds anything and you should consider deleting it.

"With an increased population aging amongst PLWH" Consider re-writing in clearer English

Methods:

Page 13, line 95: "develop" should be "developed"

Line 151: missing a space between sentences

Line 163: remove the word "the"

Line 170: I don't understand this sentence "three studies, which were the majority"--three is not the majority of 7.

I find it very confusing to combine the pediatric data with the adult data, since you would expect VERY different prevalences of ECG abnormalities among these populations. I would consider breaking this into two separate tables with separate summary statements.

Discussion:

I think the discussion is missing several points and needs to be heavily re-written. Most of the Discussion consists of the authors citing random studies from other settings but not really synthesizing the data the presented in their results in a meaningful way.

-You report a HUGE range of ECG abnormalities (6-81%) which is so large that the data is almost unhelpful. Please comment on why you think this range is so large with specific reference to study design, study population, etc. Since the studies evaluated different sets of ECG abnormalities, I don't think it's helpful to report this summary range because each study reported different kinds of ECG abnormalities.

-I think separating out the adult and pediatric studies would be more helpful for your discussion of the adults since the findings and clinical implications are so different.

-I suggest you delete this sentence: "In a prospective cohort study involving 1,770 HIV-infected and 3350 HIV negative

participants, ECG abnormalities were found in 35% of the HIV patients which was higher than 32.2% found in HIV negative individuals (43). This difference was statistically significant (p=0.04)." This adds nothing to your discussion, since you already reported results from a similar study in China (where this study was done). You also don't report comparative data between HIV-infected and HIV-uninfected so this is not relevant to your findings.

-When you discuss the SMART study, please specify which countries/regions this was in.

-Overall I think all of Paragraph 2 of the Discussion could be collapsed to a single sentence: "The prevalence of ECG abnormalities among ADULTS in our study <since 2="" adults="" all="" are="" cite="" in="" paragraph="" studies="" the="" you=""> was similar to what has been reported in high income settings" <then 2.="" all="" can="" cite="" in="" papers="" paragraph="" the="" you="">-Line 210, remove the word "likely"

-Lines 212 to 219, please specify which country each of these cited studies were in.

-I suggest deleting lines 220-229, as these are redundant from the introduction.

-"ECG are easily accessible to most settings in SSA and could play a huge role in identify PLWH at a risk of developing severe CVDs if used routinely." I think both parts of this sentence are untrue. ECGs are certainly not widely available to people with HIV in a lot of parts of SSA. I don't think ECGs could identify those at risk of developing CVDs--more accurate to say that they could identify some forms of CVD such as arryhthmias and ischemia?

-"Furthermore, it may play an important role in determining the prognosis of PLWH with suspected cardiovascular diseases." This statement is untrue.

-Again, I think a separate paragraph about pediatric results would be helpful.

-Line 234, re-write as "which may have led to exclusion of other studies reporting ECG abnormalities among persons with HIV in SSA"

-Line 235, re-write as "which likely resulted in significant heterogeneity in results"

-Line 238: I don't think it's fair to say this review presents an "exact burden" as the ranges are so large and there is so much heterogeneity among studies.

-Please re-write all three Conclusion sentences as the written English is hard to follow.</then></since>

Reviewer #2: 1. This is an important study that highlights a significant topic in which there is a large paucity of data in SSA. The manuscript will contribute to the growing body of cardiovascular disease literature in SSA and highlights the burden of disease and need for further action.

2. While the study appears to be sound, the language is unclear in some areas, making the manuscript difficult to follow at times. I would advise close review of the manuscript to ensure clarity and accuracy in the writing. I have included few lines below for some suggested changes for clarity.

Line 65 – would rephrase ‘78% of PLWH could have developed a CVD’ to ‘78% of PLWH may develop CVD’.

Line 76 – would change ‘such as arrhythmias, myocardial…’ to ‘arrhythmias and myocardial…’

Line 79-81 – would rephrase ‘With an increased population aging amongst PLWH, the prevalence of ECG abnormalities…’ to ‘With an increasingly aging population, the prevalence of ECG abnormalities among PLWH is likely to increase…’.

Line 137 – would change ‘Rate’ to lower case ‘rate’

Line 149 – would rephrase ‘The studies were done in 7 African countries…’ to ‘The studies were conducted in 7 African countries’.

In general, would rephrase the term ‘commonest’ to ‘most common’ (lines 158, 167)

Lines 170-171 – would rephrase ‘3 (43%) studies, which were the majority, reported ST elevation whose prevalence was 1.7%...’ to ‘3 (43%) studies, reported ST elevation with prevalence between 1.7%...’

Lines 206-210 are unclear and require rephrasing.

Line 230 – would change ‘huge role in identify PLWH’ to ‘huge role in identifying PLWH’

3. HIV-related characteristics of samples from the 17 studies are not described including duration of HIV, stage of HIV (i.e. well controlled/virologically suppressed vs advanced/untreated), current CD4, viral load (if available), if on ART, the type/class of ART, and the duration of ART. In addition, HIV-related characteristics were not taken into consideration in either the analysis or in the discussion. As HIV-related characteristics may be directly correlated with ECG abnormalities this is a critical piece that is missing.

4. The authors do not comment on the very wide range of ECG abnormalities found and highlight only the higher range of prevalence. I would suggest commenting on the wide range in the discussion including thoughts about why such a wide range was found, and also discussing how the wide range impacts accuracy and the interpretation of findings.

5. It is interesting that the authors include ECG abnormalities for both children and adults. As these populations are inherently different and likely have different types of cardiovascular disease, it might be interesting to describe if there were any major differences found between children and adults.

6. The authors conclude that there should be increased efforts at including ECG assessment for PLWH in SSA, however, there is no discussion regarding the multiple structural complexities that may prevent wide-scale implementation of ECG screening. I would suggest including in the discussion the logistic and structural barriers regarding the use of ECGs in SSA. For example, are ECG machines really widely available in all settings? Can patients afford the cost of an ECG if an ECG machine is available? Are there health care workers that are readily available and trained to read and interpret ECGs? If an ECG abnormality is found, are health care providers trained regarding next steps in management and are the resources available for providing those next steps in treatment?

7. In the tables I suggest changing the term HAART to ART and would also not use the term ‘positives’ and instead rephrase to PLWH or persons with HIV. In addition, would consider use of abbreviations in the table with a legend/foot note. It is also unclear to me how the studies are organized in the tables which makes it somewhat difficult to follow. It might be helpful to organize the studies in a systematic manner, and would consider organizing the studies by type of ECG abnormality or perhaps by country/region.

8. The studies by Hughes J (2019) and Mayosi BM (2006) appear to describe ECG abnormalities among persons with MDR TB and TB pericarditis. The study by Berhe N (2001) describes ECG findings for patients with visceral leishmaniasis. The ECG abnormalities from these studies may be directly related to the severity of their underlying critical illness (TB/leishmaniasis) and may not be associated with HIV. I would suggest commenting on this in the discussion or including this as a limitation.

6. PLOS authors have the option to publish the peer review history of their article (what does this mean?). If published, this will include your full peer review and any attached files.

Reviewer #1: No

Reviewer #2: No

---

## [Author Response · Author response to Decision Letter 0]

9 Jan 2022

I believe the reviewers provided a fair and thorough review of your manuscript, and I encourage you to carefully consider their comments, paying particular attention to areas where the writing is unclear and addressing the heterogeneity of studies analyzed and the limitations these bring (such as including children and adults, and including studies of patients with TB pericarditis). I agree with the reviewers that the range of ECG abnormalities is so large that it is difficult to understand their overall significance. 

 In addition to thoroughly addressing each of the comments of the reviewers, I encourage the authors to focus on which abnormalities are considered to be most clinically significant.

Thank you for the comment. clinical significance will depend on whether the ECG finding is accompanied by clinical symptoms. Most of the studies, however, did not report symptoms which accompanied the ECG results. This made it difficult for us to choose with abnormality to consider clinically relevant.

We are thankful to the reviewers and editors for the constructive comments that helped us to considerably improve our manuscript. We have carefully revised the manuscript considering the suggestions and replied to each of the concerns below. We hope that our responses satisfy the reviewers.

Reviewer #1: Overall this is a clear and well-written manuscript synthesizing data about a problem that has been under-studied. Here are my minor comments:

Introduction:

Comments:

"and is highly predictive of incident CVDs (9, 12)"--I do not think this statement is true or justified by the references. Consider rewording--"and can be predictive of incident CVDs"

"Moreover, ECG abnormalities are highly prevalent in the general population and HIV population with the

prevalence of ECG abnormalities ranging from 1.04% to 51% (9, 13-18)." I do not understand this sentence--you seem to be citing several articles about HIV-infected and HIV-uninfected patients in high-income and low-income settings. Overall, I do not think this sentence adds anything and you should consider deleting it.

"With an increased population aging amongst PLWH" Consider re-writing in clearer English

Response:

Thank you for the comment. These have been rectified.

Methods:

Comments

Page 13, line 95: "develop" should be "developed"

Line 151: missing a space between sentences

Line 163: remove the word "the"

Line 170: I don't understand this sentence "three studies, which were the majority"--three is not the majority of 7.

Response:

Thank you for the comments. These have been rectified.

Comment:

I find it very confusing to combine the pediatric data with the adult data, since you would expect VERY different prevalences of ECG abnormalities among these populations. I would consider breaking this into two separate tables with separate summary statements.

Response:

Thank you for the comment. A separate table for pediatric data has been created.

Discussion:

Comments:

I think the discussion is missing several points and needs to be heavily re-written. Most of the Discussion consists of the authors citing random studies from other settings but not really synthesizing the data the presented in their results in a meaningful way.

-You report a HUGE range of ECG abnormalities (6-81%) which is so large that the data is almost unhelpful. Please comment on why you think this range is so large with specific reference to study design, study population, etc. Since the studies evaluated different sets of ECG abnormalities, I don't think it's helpful to report this summary range because each study reported different kinds of ECG abnormalities.

Response: Thank you for comment. An explanation for the wide range has been added in line 219 to 243.

Comment:

-I think separating out the adult and pediatric studies would be more helpful for your discussion of the adults since the findings and clinical implications are so different.

Response: Thank you for the comment. Different tables have been created. Table 2 and Table 4 to cover the pediatric population.

Comments:

-I suggest you delete this sentence: "In a prospective cohort study involving 1,770 HIV-infected and 3350 HIV negative

participants, ECG abnormalities were found in 35% of the HIV patients which was higher than 32.2% found in HIV negative individuals (43). This difference was statistically significant (p=0.04)." This adds nothing to your discussion, since you already reported results from a similar study in China (where this study was done). You also don't report comparative data between HIV-infected and HIV-uninfected so this is not relevant to your findings.

-When you discuss the SMART study, please specify which countries/regions this was in.

-Overall I think all of Paragraph 2 of the Discussion could be collapsed to a single sentence: "The prevalence of ECG abnormalities among ADULTS in our study was similar to what has been reported in high income settings" -Line 210, remove the word "likely"

-Lines 212 to 219, please specify which country each of these cited studies were in.

-I suggest deleting lines 220-229, as these are redundant from the introduction.

-"ECG are easily accessible to most settings in SSA and could play a huge role in identify PLWH at a risk of developing severe CVDs if used routinely." I think both parts of this sentence are untrue. ECGs are certainly not widely available to people with HIV in a lot of parts of SSA. I don't think ECGs could identify those at risk of developing CVDs--more accurate to say that they could identify some forms of CVD such as arrhythmias and ischemia?

-"Furthermore, it may play an important role in determining the prognosis of PLWH with suspected cardiovascular diseases." This statement is untrue.

-Again, I think a separate paragraph about pediatric results would be helpful.

-Line 234, re-write as "which may have led to exclusion of other studies reporting ECG abnormalities among persons with HIV in SSA"

-Line 235, re-write as "which likely resulted in significant heterogeneity in results"

-Line 238: I don't think it's fair to say this review presents an "exact burden" as the ranges are so large and there is so much heterogeneity among studies.

Response: Thank you for the comments. These have been updated.

Comment:

-Please re-write all three Conclusion sentences as the written English is hard to follow.

Response: Thank you for the comment. The conclusion has been re-written.

Reviewer #2: 

This is an important study that highlights a significant topic in which there is a large paucity of data in SSA. The manuscript will contribute to the growing body of cardiovascular disease literature in SSA and highlights the burden of disease and need for further action.

2. While the study appears to be sound, the language is unclear in some areas, making the manuscript difficult to follow at times. I would advise close review of the manuscript to ensure clarity and accuracy in the writing. I have included few lines below for some suggested changes for clarity.

Comments

Line 65 – would rephrase ‘78% of PLWH could have developed a CVD’ to ‘78% of PLWH may develop CVD’.

Line 76 – would change ‘such as arrhythmias, myocardial…’ to ‘arrhythmias and myocardial…’

Line 79-81 – would rephrase ‘With an increased population aging amongst PLWH, the prevalence of ECG abnormalities…’ to ‘With an increasingly aging population, the prevalence of ECG abnormalities among PLWH is likely to increase…’.

Line 137 – would change ‘Rate’ to lower case ‘rate’

Line 149 – would rephrase ‘The studies were done in 7 African countries…’ to ‘The studies were conducted in 7 African countries’.

In general, would rephrase the term ‘commonest’ to ‘most common’ (lines 158, 167)

Lines 170-171 – would rephrase ‘3 (43%) studies, which were the majority, reported ST elevation whose prevalence was 1.7%...’ to ‘3 (43%) studies, reported ST elevation with prevalence between 1.7%...’

Lines 206-210 are unclear and require rephrasing.

Line 230 – would change ‘huge role in identify PLWH’ to ‘huge role in identifying PLWH’

Response: Thank you for the comment. These have been updated.

Comment:

3. HIV-related characteristics of samples from the 17 studies are not described including duration of HIV, stage of HIV (i.e. well controlled/virologically suppressed vs advanced/untreated), current CD4, viral load (if available), if on ART, the type/class of ART, and the duration of ART. In addition, HIV-related characteristics were not taken into consideration in either the analysis or in the discussion. As HIV-related characteristics may be directly correlated with ECG abnormalities this is a critical piece that is missing.

Response: Thank you for the comment. These sections have been added to the tables.

4. The authors do not comment on the very wide range of ECG abnormalities found and highlight only the higher range of prevalence. I would suggest commenting on the wide range in the discussion including thoughts about why such a wide range was found, and also discussing how the wide range impacts accuracy and the interpretation of findings.

Response: Thank you for comment. An explanation for the wide range has been added in line 219 to 243.

5. It is interesting that the authors include ECG abnormalities for both children and adults. As these populations are inherently different and likely have different types of cardiovascular disease, it might be interesting to describe if there were any major differences found between children and adults.

Response: Thank you for the comment. Different tables have been created. Table 2 and Table 4 to cover the pediatric population.

6. The authors conclude that there should be increased efforts at including ECG assessment for PLWH in SSA, however, there is no discussion regarding the multiple structural complexities that may prevent wide-scale implementation of ECG screening. I would suggest including in the discussion the logistic and structural barriers regarding the use of ECGs in SSA. For example, are ECG machines really widely available in all settings? Can patients afford the cost of an ECG if an ECG machine is available? Are there health care workers that are readily available and trained to read and interpret ECGs? If an ECG abnormality is found, are health care providers trained regarding next steps in management and are the resources available for providing those next steps in treatment?

Response: Thank you for the comment. This has been updated.

7. In the tables I suggest changing the term HAART to ART and would also not use the term ‘positives’ and instead rephrase to PLWH or persons with HIV. In addition, would consider use of abbreviations in the table with a legend/foot note. It is also unclear to me how the studies are organized in the tables which makes it somewhat difficult to follow. It might be helpful to organize the studies in a systematic manner and would consider organizing the studies by type of ECG abnormality or perhaps by country/region.

Response: Thank you for the comment. This has been updated.

8. The studies by Hughes J (2019) and Mayosi BM (2006) appear to describe ECG abnormalities among persons with MDR TB and TB pericarditis. The study by Berhe N (2001) describes ECG findings for patients with visceral leishmaniasis. The ECG abnormalities from these studies may be directly related to the severity of their underlying critical illness (TB/leishmaniasis) and may not be associated with HIV. I would suggest commenting on this in the discussion or including this as a limitation.

Response: Thank you for the comment. This has been included in the limitations line 321- 324.

---

## [Decision Letter · Decision Letter 1]

6 May 2022

PONE-D-21-34132R1PREVALENCE OF ELECTROCARDIOGRAPHIC ABNORMALITIES AMONG PEOPLE LIVING WITH HIV IN SUB-SAHARAN AFRICA: A SYSTEMATIC REVIEWPLOS ONE

Dear Dr. Semulimi,

Thank you for submitting your manuscript to PLOS ONE. After careful consideration, we feel that it has merit but still does not fully meet PLOS ONE’s publication criteria as it currently stands. Therefore, we invite you to submit a revised version of the manuscript that addresses the points raised during the review process.

The reviewers have provided additional feedback that can strengthen this manuscript further. Please note in particular the suggestion to emphasize  summary descriptions  over precise prevalence values and the request to address the clinical significance of more common ECG abnormalities. 

We look forward to receiving your revised manuscript.

Kind regards,

Nathan Maclyn Thielman, MD, MPH

Academic Editor

PLOS ONE

Reviewers' comments:

Reviewer's Responses to Questions

**Comments to the Author**

1. If the authors have adequately addressed your comments raised in a previous round of review and you feel that this manuscript is now acceptable for publication, you may indicate that here to bypass the “Comments to the Author” section, enter your conflict of interest statement in the “Confidential to Editor” section, and submit your "Accept" recommendation.

Reviewer #2: (No Response)

Reviewer #3: All comments have been addressed

2. Is the manuscript technically sound, and do the data support the conclusions?

Reviewer #2: Yes

Reviewer #3: Partly

3. Has the statistical analysis been performed appropriately and rigorously? 

Reviewer #2: Yes

Reviewer #3: N/A

4. Have the authors made all data underlying the findings in their manuscript fully available?

Reviewer #2: Yes

Reviewer #3: Yes

5. Is the manuscript presented in an intelligible fashion and written in standard English?

Reviewer #2: Yes

Reviewer #3: Yes

6. Review Comments to the Author

Reviewer #2: Overall, I think this paper is improved, and the authors are highlighting a very important topic that will significantly contribute to the cardiovascular literature in PLWH in SSA. There are some minor changes needed for clarity in language and writing in the abstract, methods, and results section. I do think the discussion needs to be heavily re-written in many areas and that perhaps there should be more of an emphasis on the clinical significance and implications of the findings.

I have included some minor comments for clarity in language and writing below.

Line 43 – I would delete (n-13), this is unnecessary as ‘Thirteen studies’ is stated immediately prior

Line 44 – I would delete (n=4) for same reasoning as above

Line 56-57 – ECG is not part of ‘risk assessment’ for CVD, but rather a tool to determine existing cardiac abnormalities. Perhaps consider changing this sentence to something like this: ‘Consideration of ECG in the comprehensive evaluation for cardiac dysfunction in PLWH in SSA may be warranted.’

Line 101 – Remove the comma after the word Science

Line 121 – ‘(Figure 1)’ should be included before the period - i.e. ‘included in the review (Figure 1).’

Line 125 – Change ‘that arose were’ to ‘that arose was…’

Line 143 – Change ‘ischemia/ Infarction’ to ‘ischemia/infarction’

Line 150 – There should be a space between Figure and 1

Line 150 – I would remove (n=4)

Lines 159-161 – These sentences are confusing and need clarification. I am uncertain why both the mean and the median age of pediatric participants are reported. Since the mean age of adults is reported, it is reasonable to also report the mean age of pediatric participants, but there is no indication to report the median as well. In addition, I assume (6.3) and (6.6) are the standard error? However, the standard error for pediatric participants is reported differently with a ‘+/-‘. Consistency and clarification are needed in reporting these measures. Also, based on the wording of these sentences, I am uncertain what Table 1 and Table 2 show. I would make this clearer somewhere in this paragraph (i.e. Table 1 describes studies reporting ECG abnormalities in adults)

Line 163 – Remove the space after the number 3

Line 165 – There is no number for the table

Lines 170–193 – I am uncertain why the number of studies is repeated after it is written out throughout the results section. For example, in Line 171, ‘Nine (9)’ and in line 174 ‘five (5)’. Was this meant to be the percentage instead of repeating the numbers again?

Lines 195-197 – This sentence is a bit unclear. Consider rewording to something like this: ‘Rate abnormalities were the most commonly reported ECG abnormalities (n=X), followed by atrial and ventricular enlargement (n=x), and lastly conduction abnormalities (n=x)

Line 197 – I am uncertain what Table 4 shows. Perhaps consider stating in the beginning of this paragraph, ‘Table 4 describes…’

Line 199 – Ventricular should have a lower case v

Lines 201-202 – There should be a comma after prolongation and another comma after 18%

Line 208 – There should be a % sign directly after 26.5, and the % after the word children should be deleted

Discussion

The discussion needs to be re-written in some areas. While there is now a discussion re: the reason for such a wide range in prevalence of abnormalities, I think that there could be more emphasis in the discussion re: the clinical significance of some of the most common ECG abnormalities. Perhaps focusing less on the overall prevalence and instead focusing more on the clinical significance and real-world implications of some of the major ECG abnormalities in the discussion would be more meaningful.

I’ve included a few specific comments for each of the paragraphs in the discussion below.

I am uncertain what the takeaway message is and am uncertain what the significance of the findings are after reading the first paragraph.

In the second paragraph – I think India is considered an LMIC and I would consider deleting that sentence.

The third paragraph is disorganized and difficult to follow. I think the takeaway of the third paragraph is that the wide range in prevalence seen in the ECG abnormalities are due to the significant differences between study samples and discrepancies in ECG interpretation in each study. Perhaps finding a way of stating this more directly and making these points more clear instead of citing and describing the characteristics of each study would be helpful.

Paragraphs 5 and 6 discussing atrial and ventricular enlargement and rate abnormalities are also disorganized and difficult to follow. I might consider focusing on the clinical significance and implications of these findings in these paragraphs.

Paragraph 7 – I don’t think ECGs are used to identified people who may be at risk for developing CVDs in the future, but instead are used to diagnose existing cardiac abnormality and dysfunction.

Tables

I find some of the tables difficult to read.

I think the titles of the tables need to be revised. For example, instead of the title ‘A table showing the characteristic of the adult review studies’ – perhaps consider changing to ‘studies reporting ECG abnormalities in adult PLWH in SSA’

In Table 1 and 2 – I’m uncertain what the p-values included in the column reporting age signifies. I also don’t think the column for HIV diagnosis in Table 1 adds anything and I’d consider deleting this. The column for ECG criteria is largely blank – and for one study the authors state ‘a standard criteria is used’. I’m uncertain what the standard criteria for ECG interpretation is. Are these left blank because the studies did not report how they interpreted the ECG? If so, I would consider deleting this column and reporting this in the results and also including this in the discussion section when discussing reasons for heterogeneity.

In Tables 3 and 4 – I am also uncertain what the p-values included signify.

Reviewer #3: It was pleasure reviewing the manuscript REVALENCE OF ELECTROCARDIOGRAPHIC ABNORMALITIES AMONG PEOPLE LIVING WITH HIV IN SUB-SAHARAN AFRICA: A SYSTEMATIC REVIEW

Summary comments

This is a welcome study that summarizes a growing body of evidence on the unique electrocardiographic characteristics among patients with HIV. The authors responses to prior comments are well noted including highlighting limitations of the study, separation of pediatric studies from adult and description of baseline characteristics of patients enrolled in the studies of interest. The language and phrasing is improved and would recommend the following minor changes.

1. Title: suggest re-framing title to “review of” rather than “prevalence of” electrographic abnormalities among people …..

2. Abstract: concise; would include the total number of retrieved studies and a statement on the limitations of the study in the abstract.

3. Introduction: well focused

4. Methods: appropriate description

5. Results: good descriptions;

- Given the small number of studies, it may be worthwhile reporting in the text, the commonest definition used for each type of abnormality considered eg criteria for LVH, basis for QTc estimations, which ST segment changes were considered ( elevation only, depression, t-wave inversion etc) between studies.

6. Discussion

- Would reframe as a review of estimates rather than precise prevalence since this was not a meta- analysis at individual patient level bur rather summary descriptions of a collection of studies. For example, would rephrase opening statement as “….our systematic review provides a comprehensive review of published ECG abnormalities among PLWH. In these studies ECG abnormalities ranged between 6.7% -81%.”

7. Limitations

- Appropriate

8. Conclusion

- Concise; Would de-emphasize prevalence and

7. PLOS authors have the option to publish the peer review history of their article (what does this mean?). If published, this will include your full peer review and any attached files.

Reviewer #2: No

Reviewer #3: No

---

## [Author Response · Author response to Decision Letter 1]

10 Jun 2022

To the Academic Editor,

PLOS One

Response to the reviewers’ comments.

Reviewer #2: Overall, I think this paper is improved, and the authors are highlighting a very important topic that will significantly contribute to the cardiovascular literature in PLWH in SSA. There are some minor changes needed for clarity in language and writing in the abstract, methods, and results section. I do think the discussion needs to be heavily re-written in many areas and that perhaps there should be more of an emphasis on the clinical significance and implications of the findings.

Answer: Dear Reviewer, thank you for the comments. The authors have revised the manuscript and changes have been made with more emphasis put on the clinical significance and implications of ECG abnormalities. 

I have included some minor comments for clarity in language and writing below.

Line 43 – I would delete (n-13), this is unnecessary as ‘Thirteen studies’ is stated immediately prior

Thank you for the comments. The comment has been addressed and the manuscript updated.

Line 44 – I would delete (n=4) for same reasoning as above

Thank you for the comments. The comment has been addressed and the manuscript updated.

Line 56-57 – ECG is not part of ‘risk assessment’ for CVD, but rather a tool to determine existing cardiac abnormalities. Perhaps consider changing this sentence to something like this: ‘Consideration of ECG in the comprehensive evaluation for cardiac dysfunction in PLWH in SSA may be warranted.’

Thank you for the comments. The comment has been addressed and the manuscript updated.

Line 101 – Remove the comma after the word Science

Thank you for the comments. The comment has been addressed and the manuscript updated.

Line 121 – ‘(Figure 1)’ should be included before the period - i.e. ‘included in the review (Figure 1).’

Thank you for the comments. The comment has been addressed and the manuscript updated.

Line 125 – Change ‘that arose were’ to ‘that arose was…’

Thank you for the comments. The comment has been addressed and the manuscript updated.

Line 143 – Change ‘ischemia/ Infarction’ to ‘ischemia/infarction’

Thank you for the comments. The comment has been addressed and the manuscript updated.

Line 150 – There should be a space between Figure and 1

Thank you for the comments. The comment has been addressed and the manuscript updated.

Line 150 – I would remove (n=4)

Answer: Thank you for the comments. The comment has been addressed and the manuscript updated.

Lines 159-161 – These sentences are confusing and need clarification. I am uncertain why both the mean and the median age of pediatric participants are reported. Since the mean age of adults is reported, it is reasonable to also report the mean age of pediatric participants, but there is no indication to report the median as well. In addition, I assume (6.3) and (6.6) are the standard error? However, the standard error for pediatric participants is reported differently with a ‘+/-‘. Consistency and clarification are needed in reporting these measures. Also, based on the wording of these sentences, I am uncertain what Table 1 and Table 2 show. I would make this clearer somewhere in this paragraph (i.e. Table 1 describes studies reporting ECG abnormalities in adults)

Thank you for the comment. The changes have been made, see below:

“The mean age of the adult participants ranged between 32±10.5 and 48±13.1 years (Table 1: Studies reporting ECG abnormalities in adult PLWH in SSA) while that of the pediatric participants was between 8.30±3.92 and 8.41±3.99 years (Table 2: Studies reporting ECG abnormalities in pediatric PLWH in SSA).”

Line 163 – Remove the space after the number 3

Thank you for the comment. The comment has been addressed and the manuscript updated.

Line 165 – There is no number for the table

Thank you for the comments. The sentence has been deleted.

Lines 170–193 – I am uncertain why the number of studies is repeated after it is written out throughout the results section. For example, in Line 171, ‘Nine (9)’ and in line 174 ‘five (5)’. Was this meant to be the percentage instead of repeating the numbers again?

Lines 195-197 – This sentence is a bit unclear. Consider rewording to something like this: ‘Rate abnormalities were the most commonly reported ECG abnormalities (n=X), followed by atrial and ventricular enlargement (n=x), and lastly conduction abnormalities (n=x)

Thank you for the comment. This has been adjusted, read below:

“followed by atrial and ventricular enlargement (n=3), and lastly conduction abnormalities (n=3)” 

Line 197 – I am uncertain what Table 4 shows. Perhaps consider stating in the beginning of this paragraph, ‘Table 4 describes…’

Thank you for the comment. The comment has been addressed and the manuscript updated.

Line 199 – Ventricular should have a lower-case v

Thank you for the comment. The comment has been addressed and the manuscript updated.

Lines 201-202 – There should be a comma after prolongation and another comma after 18%

Thank you for the comment. The comment has been addressed and the manuscript updated.

Line 208 – There should be a % sign directly after 26.5, and the % after the word children should be deleted.

Thank you for the comment. The comment has been addressed and the manuscript updated.

Discussion

The discussion needs to be re-written in some areas. While there is now a discussion re: the reason for such a wide range in prevalence of abnormalities, I think that there could be more emphasis in the discussion re: the clinical significance of some of the most common ECG abnormalities. Perhaps focusing less on the overall prevalence and instead focusing more on the clinical significance and real-world implications of some of the major ECG abnormalities in the discussion would be more meaningful.

I’ve included a few specific comments for each of the paragraphs in the discussion below.

I am uncertain what the takeaway message is and am uncertain what the significance of the findings are after reading the first paragraph.

Thank you for the message: The aim of the paragraph was to highlight the key findings from our review.

In the second paragraph – I think India is considered an LMIC and I would consider deleting that sentence.

Thank you for the comment. The comment has been addressed and the manuscript updated.

The third paragraph is disorganized and difficult to follow. I think the takeaway of the third paragraph is that the wide range in prevalence seen in the ECG abnormalities are due to the significant differences between study samples and discrepancies in ECG interpretation in each study. Perhaps finding a way of stating this more directly and making these points clearer instead of citing and describing the characteristics of each study would be helpful.

The third paragraph has been edited and reads as below:

“The wide range of ECG abnormalities could be attributed to the heterogeneity of the results and the differences in the study population. Regarding the discrepancies in the ECG reporting, several criteria such as American Heart Association Standard criteria (38), normal ECG standard for infants and children (39) and Minnesota Coding system for ECG classification (40) were used by the different studies which may have influenced the prevalence reported. In addition, the characteristics of participants included in the reviewed studies could have contributed to the reported wide range. For instance, Okoye and colleagues (31) studied PLWH who were ART naïve, aged 15 years and above with 30% of their study participants having a low CD4 count (<200 cells/µl) while in a cross-sectional study conducted in Ghana (21), most of the participants had stage III severity of disease with more than half of them (194, 56.3%) having >1 CVD risk. Among the studies conducted in the pediatric population, factors such as wasting (25) and diseases severity (28) influenced results contributing to a wider range in the prevalence of ECG abnormalities. Therefore, the absence of a uniform and standardized tool that can be adjusted for patient characteristics calls for the development of such tools to classify ECG abnormalities. This could inform risk stratification of PLWH at a risk of developing CVDs.”

Paragraphs 5 and 6 discussing atrial and ventricular enlargement and rate abnormalities are also disorganized and difficult to follow. I might consider focusing on the clinical significance and implications of these findings in these paragraphs.

The fifth paragraph has been edited and reads as below:

“Regarding atrial and ventricular enlargement in adults, findings from our study are similar to those of a cross-sectional study done in China, which reported a prevalence of left ventricular hypertrophy of 6.8% (13). In children, left ventricular hypertrophy was the second most common ECG abnormality. HIV and opportunistic infections coupled with other risk factors of CVDs such as smoking cause inflammation and release of cytokines which have been shown to predispose PLWH to atrial and ventricular hypertrophy (48). This increases the likelihood of individuals to developing ventricular and atrial arrhythmias (49), congestive heart failure and ischemic heart disease (50) which significantly contribute to CVD related mortality and morbidity.”

The sixth paragraph has been edited and reads as below:

“ECG abnormality among children unlike in the adult population. This was lower than what was reported in the United States of America (USA) (51) which was conducted before the introduction of combined ART. The higher prevalence in the USA study could be attributed to the increased risk of children to opportunistic infections subsequently leading to CVDs (52). HIV instigates viral induced myocarditis which is one of the drivers of sinus tachycardia (53) among PLWH. Sinus tachycardia predisposes patients to life threatening arrhythmias, sudden cardiac death and myocardial infarction and its presence necessitates frequent cardiac monitoring.”

“From the reviewed studies, atrial fibrillation is the most reported arrhythmia which is concordant with results from other large epidemiological studies (13, 54, 55). Studies have shown a relationship between HIV and atrial fibrillation although the underlying mechanism has not been fully elucidated (56). Atrial fibrillation causes disruptions in atrial filling and emptying leading to poor clinical outcomes such as cerebrovascular accidents, decompensated congestive cardiac failure, ischemia and sudden cardiac death causing significant morbidity and mortality in this population (29). This highlights how essential ECGs are in diagnosing PLWH with existing cardiac abnormality and dysfunction if used routinely. However, in SSA, the routine use of PLWH to ECGs is marred by the cost and the need to train health workers to read and analyse them (57).”

Paragraph 7 – I don’t think ECGs are used to identified people who may be at risk for developing CVDs in the future, but instead are used to diagnose existing cardiac abnormality and dysfunction.

Thank you for the comments. This has been updated.

Tables

I find some of the tables difficult to read.

I think the titles of the tables need to be revised. For example, instead of the title ‘A table showing the characteristic of the adult review studies’ – perhaps consider changing to ‘studies reporting ECG abnormalities in adult PLWH in SSA’

Thank you for the comment: The table title has been changed.

In Table 1 and 2 – I’m uncertain what the p-values included in the column reporting age signifies. I also don’t think the column for HIV diagnosis in Table 1 adds anything and I’d consider deleting this. The column for ECG criteria is largely blank – and for one study the authors state ‘a standard criteria is used’. I’m uncertain what the standard criteria for ECG interpretation is. Are these left blank because the studies did not report how they interpreted the ECG? If so, I would consider deleting this column and reporting this in the results and also including this in the discussion section when discussing reasons for heterogeneity.

Thank you for the comment. The column on ECG criteria has been removed and the criteria used have been discussed in the discussion.

In Tables 3 and 4 – I am also uncertain what the p-values included signify.

Thank you for the comment. The p-values have been deleted.

Reviewer #3: It was pleasure reviewing the manuscript REVALENCE OF ELECTROCARDIOGRAPHIC ABNORMALITIES AMONG PEOPLE LIVING WITH HIV IN SUB-SAHARAN AFRICA: A SYSTEMATIC REVIEW

Summary comments

This is a welcome study that summarizes a growing body of evidence on the unique electrocardiographic characteristics among patients with HIV. The authors responses to prior comments are well noted including highlighting limitations of the study, separation of pediatric studies from adult and description of baseline characteristics of patients enrolled in the studies of interest. The language and phrasing is improved and would recommend the following minor changes.

1. Title: suggest re-framing title to “review of” rather than “prevalence of” electrographic abnormalities among people …..

Thank you for the comment. The title of the review has been updated.

2. Abstract: concise; would include the total number of retrieved studies and a statement on the limitations of the study in the abstract.

Thank you for the comment. A statement on the limitations of the review has been included in the abstract. “The heterogeneity in results could be attributed to the absence of uniform criteria to define ECG abnormalities.” 

3. Introduction: well focused

4. Methods: appropriate description

5. Results: good descriptions;

- Given the small number of studies, it may be worthwhile reporting in the text, the commonest definition used for each type of abnormality considered eg criteria for LVH, basis for QTc estimations, which ST segment changes were considered (elevation only, depression, t-wave inversion etc) between studies.

Thank you for the comment. We have included the commonest definitions used to define ECG abnormalities in the reviewed studies. “In the reviewed studies, the most reported definition of ECG abnormalities was QTc interval prolongation defined as more than 0.44 seconds (n=4) (22, 25, 27, 33). Regarding ST changes, ST elevations were the most reported (n=4).”

6. Discussion

- Would reframe as a review of estimates rather than precise prevalence since this was not a meta- analysis at individual patient level bur rather summary descriptions of a collection of studies. For example, would rephrase opening statement as “….our systematic review provides a comprehensive review of published ECG abnormalities among PLWH. In these studies, ECG abnormalities ranged between 6.7% -81%.”

Thank you for the comment. The statement been updated.

7. Limitations

- Appropriate

8. Conclusion

- Concise; Would de-emphasize prevalence and

Thank you for the comment. The statement been updated.

---

## [Decision Letter · Decision Letter 2]

26 Jul 2022

PONE-D-21-34132R2Review of Electrocardiographic Abnormalities among People Living With HIV in Sub-Saharan Africa: A Systematic ReviewPLOS ONE

Dear Dr. Semulimi,

Thank you for submitting your manuscript to PLOS ONE. After careful consideration, we feel that it has merit but does not fully meet PLOS ONE’s publication criteria as it currently stands. Therefore, we invite you to submit a revised version of the manuscript that addresses the points raised during the review process.

We look forward to receiving your revised manuscript.

Kind regards,

Qigui Yu, M.D./Ph.D

Academic Editor

PLOS ONE

Journal Requirements:

Reviewers' comments:

Reviewer's Responses to Questions

**Comments to the Author**

1. If the authors have adequately addressed your comments raised in a previous round of review and you feel that this manuscript is now acceptable for publication, you may indicate that here to bypass the “Comments to the Author” section, enter your conflict of interest statement in the “Confidential to Editor” section, and submit your "Accept" recommendation.

Reviewer #2: (No Response)

Reviewer #3: All comments have been addressed

2. Is the manuscript technically sound, and do the data support the conclusions?

Reviewer #2: Yes

Reviewer #3: Yes

3. Has the statistical analysis been performed appropriately and rigorously? 

Reviewer #2: Yes

Reviewer #3: N/A

4. Have the authors made all data underlying the findings in their manuscript fully available?

Reviewer #2: Yes

Reviewer #3: Yes

5. Is the manuscript presented in an intelligible fashion and written in standard English?

Reviewer #2: Yes

Reviewer #3: Yes

6. Review Comments to the Author

Reviewer #2: Thank you for the opportunity to review this manuscript. Overall, this paper has been substantively improved and highlights an important body of work that will contribute to cardiovascular disease literature among PLWH in SSA, a high priority topic. The following minor changes for clarity are recommended prior to acceptance.

Abstract

Lines 50-51. The statement ‘the heterogeneity in results could be attributed to the absence of uniform criteria to define ECG abnormalities’ does not belong in the Results section. If the authors wish to keep this sentence in this abstract, it should be included in the Conclusion section of the abstract.

Line 55. The word ‘maybe’ should be ‘may be’

Methods

Line 92. The period should come after (S1 Checklist). Please revise ‘…(PRISMA) checklist (15). (S1 Checklist)’ to ‘(PRISMA) checklist (S1 Checklist) (15).’

Line 135. Same as above, the period should come after (S3 Checklist). Please revise to ‘…high risk of bias (see S3 Checklist).’

Results

Lines 153 – 156. ‘In the reviewed studies, the most reported definition of ECG abnormalities was QTc interval prolongation defined as more than 0.44 seconds (n=4) (22, 25, 27, 33). Regarding ST changes, ST elevations were the most reported (n=4).’ I think it is fine to include these new sentences, but they seem misplaced in this paragraph. Perhaps consider moving somewhere in the paragraph below to ECG Abnormalities.

Lines 157 – 161. I would not include the titles of the tables in the manuscript text. Would consider revising lines 157-161 to the following:

‘Table 1 summarizes the studies describing ECG abnormalities in adults living with HIV in SSA while Table 2 reports ECG abnormalities in pediatric participants. The mean age of the adult participants ranged between 32±10.5 and 48±13.1 years while that of the pediatric participants was between 8.30±3.92 and 8.41±3.99 years.’

Line 184. Would revise ‘ST elevation whose prevalence was 1.7%...’ to ‘ST elevation with prevalence ranging from 1.7%...’

Line 187. Would revise ‘0.4% — 20.4%’ to ‘0.4% to 20.4%’

Discussion

Line 206. Would revise ‘6.7% — 26.5% in children.’ To ‘6.7% to 26.5% in children.’

Line 213. Would revise ‘19%— 51% of participants’ to ‘19% to 51% of participants’

Line 216. Would add clarification that this sentence pertains to the authors’ study. Would consider revising sentence to: ‘The wide range of ECG abnormalities in our study could be attributed to the heterogeneity of the results and the differences in the study population.’

Lines 221 – 222. Would consider revising this sentence to clarify that diversity/heterogeneity of sample contributed to wide range of findings. Perhaps revise to something like this: ‘In addition, the diverse range of characteristics of participants included in the reviewed studies could have contributed to the reported wide range.’

Line 225. Would include a comma before the word while.

‘…cells/uL), while in a cross-sectional study…’

Lines 228. Would revise ‘diseases’ to ‘disease’

Line 228 - 229. Would revise ‘influenced results contributing to a wider range in the prevalence of ECG abnormalities’ to ‘influenced results, likely contributing to the wide range seen in the prevalence of ECG abnormalities.’

Lines 229 - 232. ECGs are not used as risk stratification for development of CVDs. Furthermore, there are standardized protocols used to analyze ECGs, however, analysis of ECGs still is dependent on the person interpreting the EKG.

Would revise the following sentences: ‘Therefore, the absence of a uniform and standardized tool that can be adjusted for patient characteristics calls for the development of such tools to classify ECG abnormalities. This could inform risk stratification of PLWH at a risk of developing CVDs.’

Perhaps revise to something like this:

‘The absence of a uniform and standardized approach in analyzing ECGs likely led to heterogeneity in results. Further studies that utilize uniform and standardized protocols in ECG interpretation are needed to better assess ECG abnormalities identified that may represent CVD dysfunction in PLWH.’

Line 237. Add a comma before the word which

Line 238. Add a comma after the word recently

The HIV effects on cardiomyocytes leading to fibrosis as well as certain classes of ART such as non-nucleoside reverse transcriptase inhibitors, which have been part of the first line regimen until recently, could have contributed to the high prevalence of conduction abnormalities and QTc prolongation (41, 46).

Lines 242 – 243. Would revise ‘Regarding atrial and ventricular enlargement in adults, findings from our study are similar to those of a cross-sectional study…’ to:

‘Findings from our study regarding atrial and ventricular enlargement in adults, are similar to those of a cross-sectional study…’

Line 251. Would revise ‘In the pediatric population, sinus tachycardia, a rate abnormality was the most common’ to:

‘In the pediatric population, rate abnormality, specifically sinus tachycardia, was the most common..’

Lines 253 – 256. Would revise the following for clarity. ‘This was lower than what was reported in the United States of America (USA) (51) which was conducted before the introduction of combined ART. The higher prevalence in the USA study could be attributed to the increased risk of children to opportunistic infections subsequently leading to CVDs (52).’

Would revise to:

‘This was lower than what has been previously reported in a study conducted in the United States (US) (51) which was conducted before the introduction of combined ART (cART). The higher prevalence of rate abnormalities previously described in the US study could be attributed to the increased risk of opportunistic infections in the absence of cART subsequently leading to CVDs (52).’

Lines 257 – 259. I don’t think this sentence re: sinus tachycardia is true. ‘Sinus tachycardia predisposes patients to life threatening arrhythmias, sudden cardiac death and myocardial infarction and its presence necessitates frequent cardiac monitoring.’

Perhaps revise to something like this:

‘Sinus tachycardia may be associated with ischemic heart disease, heart failure and other cardiovascular morbidities, and its presence necessitates evaluation of the underlying etiology and cardiac monitoring.’

Line 260. For increased clarity, would revise ‘From the reviewed studies’ to ‘In our systematic review’

Line 260. Revise ‘atrial fibrillation is the’ to ‘atrial fibrillation was the’

Line 266 – 268. Would revise this sentence for improved clarity. ‘This highlights how essential ECGs are in diagnosing PLWH with existing cardiac abnormality and dysfunction if used routinely.’

Would revise to:

‘This highlights the importance of ECGs in identifying existing cardiac abnormality and dysfunction in PLWH.’

Line 268 – 269. Would revise this sentence for increased clarity. ‘However, in SSA, the routine use of PLWH to ECGs is marred by the cost and the need to train health workers to read and analyse them (57).’

Would revise to:

‘However, in SSA, the routine use of ECGs among PLWH is hindered by multiple system and clinic level challenges such as cost and lack of trained health workers required for accurate ECG interpretation.’

Line 272. Revise ‘there is no uniform’ to ‘there was no uniform’

Lines 276. Revise for clarity ‘…leishmaniasis respectively which may have been associated or their synergistic effect with HIV could…’ to ‘…leishmaniasis and these infections, or their synergistic effects with HIV, could…’

Line 290. Revise ‘incorporate’ to ‘incorporated’

Tables

--Would include any abbreviations used in the Table as a footnote at the end of the table

--For Tables 1 and 2, perhaps in the column ‘ECG criteria used’ would consider adding a * for those studies that did not report what criteria was used, and then at footnote at the end of the table, could clarify that * means that the criteria used to interpret ECGs was not reported

--Table 3 and 4. For increased clarity would consider changing the title of Table 3 and 4 to:

ECG abnormalities among reviewed adult studies

ECG abnormalities among reviewed pediatric studies

Reviewer #3: It was a pleasure reviewing the revised manuscript: Review of Electrocardiographic Abnormalities among People Living With HIV in Sub-Saharan Africa: A Systematic Review

- No further comments;

7. PLOS authors have the option to publish the peer review history of their article (what does this mean?). If published, this will include your full peer review and any attached files.

Reviewer #2: No

Reviewer #3: No

---

## [Author Response · Author response to Decision Letter 2]

29 Jul 2022

Dear reviewers,

Thank you for your comments. Below are the responses to your comments.

Abstract

Lines 50-51. The statement ‘the heterogeneity in results could be attributed to the absence of uniform criteria to define ECG abnormalities’ does not belong in the Results section. If the authors wish to keep this sentence in this abstract, it should be included in the Conclusion section of the abstract.

This has been moved to the conclusion section, line 54-55.

Line 55. The word ‘maybe’ should be ‘may be’

This has been corrected.

Methods

Line 92. The period should come after (S1 Checklist). Please revise ‘…(PRISMA) checklist (15). (S1 Checklist)’ to ‘(PRISMA) checklist (S1 Checklist) (15).’

This has been corrected.

Line 135. Same as above, the period should come after (S3 Checklist). Please revise to ‘…high risk of bias (see S3 Checklist).’

This has been corrected.

Results

Lines 153 – 156. ‘In the reviewed studies, the most reported definition of ECG abnormalities was QTc interval prolongation defined as more than 0.44 seconds (n=4) (22, 25, 27, 33). Regarding ST changes, ST elevations were the most reported (n=4).’ I think it is fine to include these new sentences, but they seem misplaced in this paragraph. Perhaps consider moving somewhere in the paragraph below to ECG Abnormalities.

This has been moved to ECG abnormalities, line 171-173.

Lines 157 – 161. I would not include the titles of the tables in the manuscript text. Would consider revising lines 157-161 to the following:

‘Table 1 summarizes the studies describing ECG abnormalities in adults living with HIV in SSA while Table 2 reports ECG abnormalities in pediatric participants. The mean age of the adult participants ranged between 32±10.5 and 48±13.1 years while that of the pediatric participants was between 8.30±3.92 and 8.41±3.99 years.’

This has been corrected, line 158-162.

Line 184. Would revise ‘ST elevation whose prevalence was 1.7%...’ to ‘ST elevation with prevalence ranging from 1.7%...’

This has been corrected.

Line 187. Would revise ‘0.4% — 20.4%’ to ‘0.4% to 20.4%’

This has been corrected.

Discussion

Line 206. Would revise ‘6.7% — 26.5% in children.’ To ‘6.7% to 26.5% in children.’

This has been corrected.

Line 213. Would revise ‘19%— 51% of participants’ to ‘19% to 51% of participants’

This has been corrected.

Line 216. Would add clarification that this sentence pertains to the authors’ study. Would consider revising sentence to: ‘The wide range of ECG abnormalities in our study could be attributed to the heterogeneity of the results and the differences in the study population.’

This has been revised, line 224.

Lines 221 – 222. Would consider revising this sentence to clarify that diversity/heterogeneity of sample contributed to wide range of findings. Perhaps revise to something like this: ‘In addition, the diverse range of characteristics of participants included in the reviewed studies could have contributed to the reported wide range.’

This has been revised, line 229.

Line 225. Would include a comma before the word while.

‘…cells/uL), while in a cross-sectional study…’

This has been corrected.

Lines 228. Would revise ‘diseases’ to ‘disease’

This has been corrected.

Line 228 - 229. Would revise ‘influenced results contributing to a wider range in the prevalence of ECG abnormalities’ to ‘influenced results, likely contributing to the wide range seen in the prevalence of ECG abnormalities.’

This has been revised.

Lines 229 - 232. ECGs are not used as risk stratification for development of CVDs. Furthermore, there are standardized protocols used to analyze ECGs, however, analysis of ECGs still is dependent on the person interpreting the EKG.

Would revise the following sentences: ‘Therefore, the absence of a uniform and standardized tool that can be adjusted for patient characteristics calls for the development of such tools to classify ECG abnormalities. This could inform risk stratification of PLWH at a risk of developing CVDs.’

Perhaps revise to something like this:

‘The absence of a uniform and standardized approach in analyzing ECGs likely led to heterogeneity in results. Further studies that utilize uniform and standardized protocols in ECG interpretation are needed to better assess ECG abnormalities identified that may represent CVD dysfunction in PLWH.’

This has been revised, line 237 to 241.

Line 237. Add a comma before the word which

This has been corrected.

Line 238. Add a comma after the word recently

The HIV effects on cardiomyocytes leading to fibrosis as well as certain classes of ART such as non-nucleoside reverse transcriptase inhibitors, which have been part of the first line regimen until recently, could have contributed to the high prevalence of conduction abnormalities and QTc prolongation (41, 46).

This has been corrected.

Lines 242 – 243. Would revise ‘Regarding atrial and ventricular enlargement in adults, findings from our study are similar to those of a cross-sectional study…’ to:

‘Findings from our study regarding atrial and ventricular enlargement in adults, are similar to those of a cross-sectional study…’

This has been revised, line 254 to 255.

Line 251. Would revise ‘In the pediatric population, sinus tachycardia, a rate abnormality was the most common’ to:

‘In the pediatric population, rate abnormality, specifically sinus tachycardia, was the most common..’

This has been revised, line 265 to 266

Lines 253 – 256. Would revise the following for clarity. ‘This was lower than what was reported in the United States of America (USA) (51) which was conducted before the introduction of combined ART. The higher prevalence in the USA study could be attributed to the increased risk of children to opportunistic infections subsequently leading to CVDs (52).’

Would revise to:

‘This was lower than what has been previously reported in a study conducted in the United States (US) (51) which was conducted before the introduction of combined ART (cART). The higher prevalence of rate abnormalities previously described in the US study could be attributed to the increased risk of opportunistic infections in the absence of cART subsequently leading to CVDs (52).’

This has been revised, line 267-273.

Lines 257 – 259. I don’t think this sentence re: sinus tachycardia is true. ‘Sinus tachycardia predisposes patients to life threatening arrhythmias, sudden cardiac death and myocardial infarction and its presence necessitates frequent cardiac monitoring.’

Perhaps revise to something like this:

‘Sinus tachycardia may be associated with ischemic heart disease, heart failure and other cardiovascular morbidities, and its presence necessitates evaluation of the underlying etiology and cardiac monitoring.’

This has been revised, line 277 to 279.

Line 260. For increased clarity, would revise ‘From the reviewed studies’ to ‘In our systematic review’

This has been corrected.

Line 260. Revise ‘atrial fibrillation is the’ to ‘atrial fibrillation was the’

This has been corrected.

Line 266 – 268. Would revise this sentence for improved clarity. ‘This highlights how essential ECGs are in diagnosing PLWH with existing cardiac abnormality and dysfunction if used routinely.’

Would revise to:

‘This highlights the importance of ECGs in identifying existing cardiac abnormality and dysfunction in PLWH.’

This has been revised, line 289- 290.

Line 268 – 269. Would revise this sentence for increased clarity. ‘However, in SSA, the routine use of PLWH to ECGs is marred by the cost and the need to train health workers to read and analyse them (57).’

Would revise to:

‘However, in SSA, the routine use of ECGs among PLWH is hindered by multiple system and clinic level challenges such as cost and lack of trained health workers required for accurate ECG interpretation.’

This has been revised, line 292- 294.

Line 272. Revise ‘there is no uniform’ to ‘there was no uniform’

This has been revised.

Lines 276. Revise for clarity ‘…leishmaniasis respectively which may have been associated or their synergistic effect with HIV could…’ to ‘…leishmaniasis and these infections, or their synergistic effects with HIV, could…’

This has been revised.

Line 290. Revise ‘incorporate’ to ‘incorporated’

This has been corrected.

Tables

--Would include any abbreviations used in the Table as a footnote at the end of the table

The abbreviations have been included as footnotes to the tables.

--For Tables 1 and 2, perhaps in the column ‘ECG criteria used’ would consider adding a * for those studies that did not report what criteria was used, and then at footnote at the end of the table, could clarify that * means that the criteria used to interpret ECGs was not reported.

In the previous set of comments, I was asked to remove the column on ECG criteria since I had included it in the results, hence its absence in this version of the manuscript.

--Table 3 and 4. For increased clarity would consider changing the title of Table 3 and 4 to:

ECG abnormalities among reviewed adult studies

ECG abnormalities among reviewed pediatric studies

This has been corrected.

---

## [Decision Letter · Decision Letter 3]

20 Feb 2023

PONE-D-21-34132R3Review of Electrocardiographic Abnormalities among People Living With HIV in Sub-Saharan Africa: A Systematic ReviewPLOS ONE

Dear Dr. Semulimi,

Thank you for submitting your manuscript to PLOS ONE. After careful consideration, we feel that it has merit but does not fully meet PLOS ONE’s publication criteria as it currently stands. Therefore, we invite you to submit a revised version of the manuscript that addresses the points raised during the review process.

We look forward to receiving your revised manuscript.

Kind regards,

Nigusie Selomon Tibebu, MSc

Academic Editor

PLOS ONE

Journal Requirements:

Additional Editor Comments:

I have assessed your manuscript and marked it for a minor revision to inform you that, in its present form, it cannot be considered for publication as it does not meet the criteria for PLOS ONE. I discovered the following core issues: 1-How could with a similar registration number (CRD42021243664) in research Square (https://doi.org/10.21203/rs.3.rs-743165/v1)

2-What was the main significance of publishing solely systematic review (Plos One), and systematic review and Meta analysis (research square/priprint)?

Reviewers' comments:

Reviewer's Responses to Questions

**Comments to the Author**

1. If the authors have adequately addressed your comments raised in a previous round of review and you feel that this manuscript is now acceptable for publication, you may indicate that here to bypass the “Comments to the Author” section, enter your conflict of interest statement in the “Confidential to Editor” section, and submit your "Accept" recommendation.

Reviewer #3: All comments have been addressed

2. Is the manuscript technically sound, and do the data support the conclusions?

Reviewer #3: Yes

3. Has the statistical analysis been performed appropriately and rigorously? 

Reviewer #3: N/A

4. Have the authors made all data underlying the findings in their manuscript fully available?

Reviewer #3: Yes

5. Is the manuscript presented in an intelligible fashion and written in standard English?

Reviewer #3: Yes

6. Review Comments to the Author

Reviewer #3: (No Response)

7. PLOS authors have the option to publish the peer review history of their article (what does this mean?). If published, this will include your full peer review and any attached files.

Reviewer #3: No

---

## [Author Response · Author response to Decision Letter 3]

27 Feb 2023

Response to the Editor’s comments

How could with a similar registration number (CRD42021243664) in research Square (https://doi.org/10.21203/rs.3.rs-743165/v1)

Thank you for the comment. Indeed, an online article titled “Prevalence of Electrocardiographic Abnormalities Among People Living With HIV/AIDS in Sub-Saharan Africa: Protocol for a Systematic Review and Meta-analysis” with the same registration number CRD42021243664 was published in research square. However, kindly note that the said article is a pre-print which did not undergo peer review. Secondly, the article is the protocol for the systematic review and meta-analysis as it is stated clearly in the title. It is a common practise for journals to publish protocols of Systematic reviews and meta-analysis, case in point:

https://journals.plos.org/plosone/s/submission-guidelines#loc-lab-protocols

https://bmjopen.bmj.com/pages/authors

https://systematicreviewsjournal.biomedcentral.com/submission-guidelines/preparing-your-manuscript/protocol

2-What was the main significance of publishing solely systematic review (PLOS One), and systematic review and Meta analysis (research square/preprint)?

Thank you for the comments. We submitted and are hoping to publish this paper as a systematic review and not as systematic review and Meta analysis as intended in the protocol because there were no uniform criteria to define ECG abnormalities. This made the pooling of the data retrieved from the published papers difficult making it impossible for us to conduct a meta-analysis which was stated as a limitation, line 272—273. 

Journal Requirements:

Please review your reference list to ensure that it is complete and correct. If you have cited papers that have been retracted, please include the rationale for doing so in the manuscript text or remove these references and replace them with relevant current references. Any changes to the reference list should be mentioned in the rebuttal letter that accompanies your revised manuscript. If you need to cite a retracted article, indicate the article’s retracted status in the References list and also include a citation and full reference for the retraction notice.

Thank you for the comment. The reference list is up to date and no changes have been made to it.

---

## [Editor Report · Decision Letter 4]

9 Mar 2023

Review of Electrocardiographic Abnormalities among People Living With HIV in Sub-Saharan Africa: A Systematic Review

PONE-D-21-34132R4

Dear Dr. Semulimi,

We’re pleased to inform you that your manuscript has been judged scientifically suitable for publication and will be formally accepted for publication once it meets all outstanding technical requirements.

Kind regards,

Nigusie Selomon Tibebu, MSc

Academic Editor

PLOS ONE
---

## [Editor Report · Acceptance letter]

15 Mar 2023

PONE-D-21-34132R4 

Review of Electrocardiographic Abnormalities among People Living With HIV in Sub-Saharan Africa: A Systematic Review 

Dear Dr. Semulimi:

I'm pleased to inform you that your manuscript has been deemed suitable for publication in PLOS ONE. Congratulations! Your manuscript is now with our production department. 

Kind regards, 

on behalf of

Assistant Professor Nigusie Selomon Tibebu 

Academic Editor

PLOS ONE